# Towards Thinking-Optimal Scaling of Test-Time Compute for LLM Reasoning

**Wenkai Yang**[1*]**, Shuming Ma**[2]**, Yankai Lin**[1†]**, Furu Wei**[2]
[1]Gaoling School of Artificial Intelligence, Renmin University of China
[2]Microsoft Research
{wenkaiyang, yankailin}@ruc.edu.cn
{shuming.ma, fuwei}@microsoft.com

## Abstract

Recent studies have shown that making a model spend more time thinking through longer Chain of Thoughts (CoTs) enables it to gain significant improvements in complex reasoning tasks. While current researches continue to explore the benefits of increasing test-time compute by extending the CoT lengths of Large Language Models (LLMs), we are concerned about a potential issue hidden behind the current pursuit of test-time scaling: *Would excessively scaling the CoT length actually bring adverse effects to a model's reasoning performance?* Our explorations on mathematical reasoning tasks reveal an unexpected finding that scaling with longer CoTs can indeed impair the reasoning performance of LLMs in certain domains. Moreover, we discover that there exists an optimal scaled length distribution that differs across different domains. Based on these insights, we propose a Thinking-Optimal Scaling strategy. Our method first uses a small set of seed data with varying response length distributions to teach the model to adopt different reasoning efforts for deep thinking. Then, the model selects its shortest correct response under different reasoning efforts on additional problems for self-improvement. Our self-improved models built upon Qwen2.5-32B-Instruct outperform other distillation-based 32B o1-like models across various math benchmarks, and achieve performance on par with the teacher model QwQ-32B-Preview that produces the seed data.[3]

## 1 Introduction

Recently, System-2 thinking [44] has become an important research area for enhancing the reasoning capabilities of Large Language Models (LLMs). Unlike previous System-1 thinking systems [45, 35] that perform fast thinking, such a slow thinking system aims to increase the test-time compute of LLMs to make them think more thoroughly before responding to a question. OpenAI' o1 model [28] has demonstrated a promising potential in this direction. By incentivizing the model to employ longer internal Chain of Thoughts (CoTs) [42] for thinking, o1 shows human-like reasoning capabilities, including searching, reflecting, backtracking, and re-exploring in its reasoning process, and achieves outstanding performance on complex reasoning tasks [13, 33].

Subsequently, a series of follow-up studies [30, 36, 11] have been proposed to imitate and explore o1-like thinking systems. These studies try to scale the number of reasoning tokens of LLMs either by distilling from existing o1-like models [14, 25, 26] or reinforcement learning [6, 15], and gain significant improvements compared to earlier reasoning models [45, 35].

---

[*]Work done during an internship at Microsoft Research.
[†]Corresponding Author
[3]Code, data and models are available at `https://github.com/RUCBM/TOPS`.

Behind the promising paradigm of test-time scaling, there is a few concurrent studies [3, 22] highlighting an efficiency issue of overthinking in existing o1-like models, where they tend to generate an excessive number of tokens, even for simple questions that could be answered correctly with just a few tokens. However, we are concerned about a more critical issue that *could the excessive pursuit of longer CoTs have negative impacts on the model's reasoning performance?* That is, besides the efficiency issues, we aim to explore and study whether and how overly test-time scaling could potentially impair the reasoning performance of LLMs, typically in the math domain.

To study the problem, we first calculate and compare the accuracies and used reasoning tokens of several o1-like models and their corresponding System-1 thinking models on MATH500 [20] and AIME2024[4] (see Figure 1). We find that subsequent o1-like models, QwQ-32B-Preview [30] as an typical example, generate much more tokens but gain only limited improvements in model performance. This preliminary exploration indicates that scaling to more reasoning tokens might not consistently lead to better performance. Then, to fairly investigate the performance comparison of the same base model after scaling with different lengths of CoTs, we conduct additional experiments on LLaMA3.1-8B-Instruct [24] and Qwen2.5-32B-Instruct [29]. Specifically, we utilize QwQ-32B-Preview [30] to generate and filter three types of reasoning paths with different lengths for the same set of prompts. Then, we teach the base model to use different reasoning efforts (i.e., different numbers of reasoning tokens) to solve a given problem based on learning on different subsets. Surprisingly, we find that training with longer reasoning paths leads to worse performance especially in easier tasks, and there exists an optimal reasoning effort that varies across tasks of different difficulty levels. Our further analysis reveals that longer CoTs may contain more erroneous steps. Though including a certain incorrect steps and subsequent reflective steps can teach the model how to correct errors in inference, training on excessive erroneous steps can have a negative impact.

Based on the above findings, we propose a Thinking-OPtimal Scaling strategy (TOPS) that allows LLMs to decide by themselves how many tokens are needed to solve a given problem. The motivation is, if an LLM can already answer a question correctly under the given reasoning effort, increasing the response length with additional tokens may have adverse effects as longer responses are more likely to include erroneous steps. On the other hand, encouraging LLMs to spend more time thinking brings benefits to tackling more challenging problems. Therefore, we first use a small set of o1-like responses under different reasoning efforts (i.e., of varying lengths) to train a "tag" model, which is used to generate responses for a large set of math problems under different reasoning efforts. Then, we select the shortest correct response generated across all reasoning efforts given the same problem to create a thinking-optimal dataset, which is used for the self-improvement of the base model. Our self-improved model based on Qwen2.5-32B-Instruct achieves better performance than existing distillation-based 32B o1-like models in various benchmarks with varying levels of difficulty, including GSM8K [4], MATH500 and AIME2024. Furthermore, we perform iterative self-improvement and obtain a reasoning model that achieves comparable performance to QwQ-32B-Preview.

## 2   Related work

**LLM Reasoning** Leveraging the Chain-of-Thought (CoT) technique [42, 51], LLMs have demonstrated impressive performance on various reasoning tasks [34, 33, 35]. CoT enables LLMs to decompose the entire problem into several sub-goals and then reason step-by-step to achieve a more reliable answer. Among various LLM reasoning tasks, mathematical reasoning has become one of the most widely studied and important tasks. Current work on LLM math reasoning primarily focuses on: synthesizing large-scale and diverse math data [49, 48, 17], constructing challenging math reasoning benchmarks [8, 9], training powerful process reward models (PRMs) [20, 39, 36], and designing more effective algorithms to improve math reasoning capabilities of LLMs [16, 12, 5].

**Test-Time Scaling** Recently, scaling test-time compute of LLMs has shown significant potential for further improving their reasoning performance [1, 2]. Existing test-time scaling studies can be divided into several categories: (1) **Sampling-based scaling** aims to increase the number of individual reasoning paths of LLMs when solving a given problem. Then the most reliable answer is selected from all the generated options using mechanisms such as majority voting [40], weighted majority voting [18], or best-of-N selection [20]. (2) **Tree search-based scaling** expands reasoning paths by

---

[4]`https://huggingface.co/datasets/AI-MO/aimo-validation-aime`

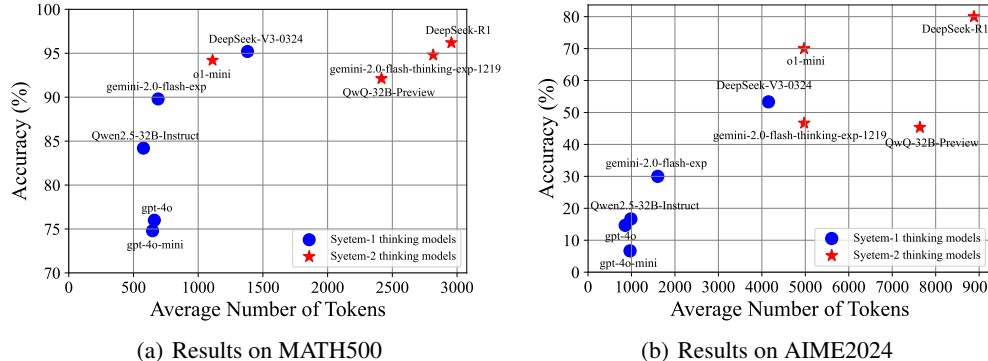

(a) Results on MATH500      (b) Results on AIME2024

Figure 1: The accuracy and the average number of tokens for each model on MATH500 and AIME2024. To ensure a fair comparison, we tokenized all model outputs using the Qwen2.5 tokenizer.

constructing tree-like trajectories, allowing LLMs to explore diverse options at each state and continue reasoning along the most promising directions. Tree-of-Thoughts (ToT) [47] and Monte Carlo Tree Search (MCTS) [43, 50, 37, 52] are two typical tree search-based test-time scaling methods. (3) **In-context search-based scaling** enables LLMs to learn to search, backtrack and re-explore within one single CoT path [7]. Recently, OpenAI's o1 model [28] have made a significant breakthrough in this line. It leverages reinforcement learning to scale the lengths of CoT to enable LLMs to perform thorough thinking through reflection, verification and re-exploration when solving problems. Following the same line, a series of studies [36, 6, 30, 11, 25, 14, 19, 46] have been proposed to scale CoT lengths during inference time. Our work also primarily focuses on the scaling properties of o1-like models.

We notice that there is a few concurrent studies [3, 22] highlighting that existing o1-like models exhibit overthinking issues, often generating an excessive number of tokens for simple problems with minimal benefit. Thus, they aim to shorten the CoT lengths of o1-like models while preserving their performance. However, our work differs in that we aim to uncover a deeper and more critical issue: scaling with more tokens can, in some cases, even degrade the model's performance. Thus, our work focuses on achieving optimal test-time scaling from base models in both aspects of effectiveness and efficiency.

## 3 The impact of scaling efforts on the effectiveness of test-time scaling

### 3.1 Preliminary analysis on existing o1-like models

Though o1-like models has proven to be much more effective on reasoning tasks than previous System-1 thinking models, we are curious about the scaling process behind the these o1-like models. That is, we want to explore that: *How effective has their scaling achieved compared to their corresponding System-1 thinking models(e.g., QwQ-32B-Preview v.s. Qwen2.5-32B-Instruct)?*

We first conduct a preliminary analysis on several existing typical o1-like models along with their corresponding System-1 thinking models. Specifically, we choose o1-mini [28] v.s. gpt-4o/4o-mini [27],[5] Gemini2.0-Flash-Thinking-Exp.-1219 [11] v.s. Gemini2.0-Flash-Exp. [10], QwQ-32B-Preview [30] v.s. Qwen2.5-32B-Instruct [29], and DeepSeek-R1 [6] v.s. DeepSeek-V3 [21] as our experimental models. We calculate the accuracy and the average number of generated tokens of each model on two typical benchmarks: **MATH500** [20]: 500 high school math competition problems across various subjects, sampled from MATH benchmark [13]; **AIME2024**: 30 challenging problems from the American Invitational Mathematics Examination (AIME). To address the issue of token counts not being directly comparable due to the different tokenizers used by different models, we standardize by using Qwen2.5 tokenizer to tokenize the reasoning completions of different models

---

[5]Note that o1-mini and 4o/4o-mini do not have equivalent number of parameters, but we make a rough comparison here.

Table 1: Data statistics (number of problems and average number of tokens in responses for each type of reasoning effort) of three types of data samples under different reasoning efforts for training each tag model.

| Model | #Problems | #Tokens | | |
| --- | --- | --- | --- | --- |
| | | Low | Medium | High |
| LLaMA3.1-8B-Tag | 1256 | 1532.32 | 2460.07 | 3647.50 |
| Qwen2.5-32B-Tag | 1312 | 1588.23 | 2535.65 | 3767.92 |

and then calculate the number of tokens. As the internal CoT of o1-mini is not available to users, we use an estimation strategy based on the summary part, the number of reasoning tokens and total number of completion tokens returned from the o1-mini model to estimate the number of tokens of hidden CoT tokenized by Qwen2.5 tokenizer. Details and further discussions are in Appendix B. We set the maximum number of generation tokens to 16,384 for each model in all evaluations.

We put the visualization results in Figure 1. As we can see, subsequent o1-like models (QwQ-32B-Preview and Gemini2.0-Flash-Thinking) show less effective scaling effects compared with o1-mini, as they generate much more tokens but gain less improvements when scaling from their corresponding System-1 thinking models. QwQ-32B-Preview has the most severe issue in this regard. This preliminary analysis suggests, to some extent, that excessively scaling to longer CoTs does not maximize test-time scaling effects.

## 3.2 Deeper explorations on the scaling process of CoT length

The above analysis still faces a problem that the base models of different o1-like models are not identical, making it unfairly to compare the impacts of scaled CoT lengths on test-time scaling effects of different models. Therefore, we conduct experiments on LLaMA3.1-8B-Instruct [24] and Qwen2.5-32B-Instruct [29] to fairly investigate the impact of the scaling efforts on the effectiveness of test-time scaling. Specifically, we first use three system prompts (refer to Figure 7), corresponding to different levels of reasoning effort ("Low", "Medium" and "High"), to prompt QwQ-32B-Preview to generate solutions of different numbers of tokens for the same set of math problems sampled from NuminaMath [17]. We then filter out the problems that can be answered correctly under all three reasoning efforts, along with the corresponding three reasoning paths of different lengths. However, we find that QwQ-32B-Preview has relatively poor instruction-following abilities, reflected in that the length distributions of the generated responses for the same question does not closely match the specified system prompts (refer to the empirical evidence in Appendix G). Therefore, for a given problem, we further reorder the three responses based on their lengths and keep them if their pairwise length difference consistently exceeds 300 tokens. The length is determined either by LLaMA3.1 tokenizer or Qwen2.5 tokenizer depending on the chosen experimental model. Finally, we curate a set of 1.3K problems, each accompanied by three o1-like responses of varying lengths. The data statistics of each set for each model is shown in Table 1. We assign different system prompts (the same in Figure 7) to each type of responses and train the base model on all three types of samples. By doing so, we ensure the consistency between the base model and the training problems, allowing for a fair comparison on the impacts of different scaling efforts on the effectiveness of test-time scaling. We refer to the fine-tuned model as the "tag" model (LLaMA3.1-8B-Tag and Qwen2.5-32B-Tag). During inference, we can use different system prompts to guide the tag model in applying varying levels of reasoning effort to answer the questions.

The detailed training settings are put in Appendix E.1. For evaluation, besides **MATH500** and **AIME2024** introduced before, we further include **GSM8K** [4] that contains 1319 grade school math word problems. In the following, unless otherwise specified, we set the decoding temperature to 1.0 for all o1-like models, by following the recommended setting for o1 model.[6] For System-1 thinking models LLaMA3.1-8B-Instruct and Qwen2.5-32B-Instruct, we report the results for both decoding temperatures with 1.0 and 0.0 for comprehensive comparison. Each result under 1.0 temperature is averaged over 5 random seeds. The full results on LLaMA3.1-8B-Instruct and Qwen2.5-32B-Instruct are shown in Figure 2 and Figure 3 respectively. We also display the performance of QwQ-32B-

---

[6]https://platform.openai.com/docs/guides/reasoning

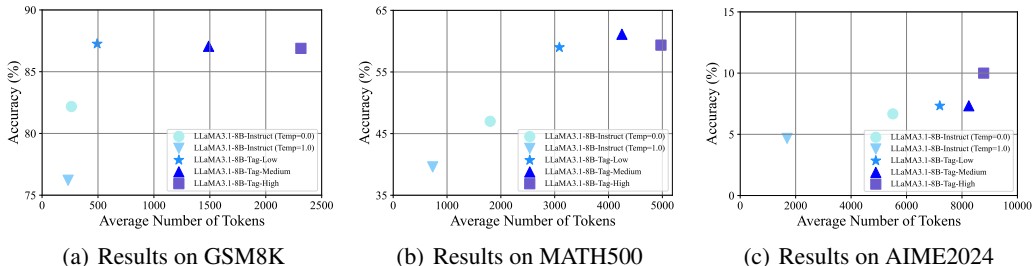

(a) Results on GSM8K  (b) Results on MATH500  (c) Results on AIME2024

Figure 2: The accuracy and the average number of tokens of LLaMA3.1-8B-Instruct and LLaMA3.1-8B-Tag under different reasoning efforts ("Low", "Medium" and "High") on different benchmarks with varying levels of difficulty.

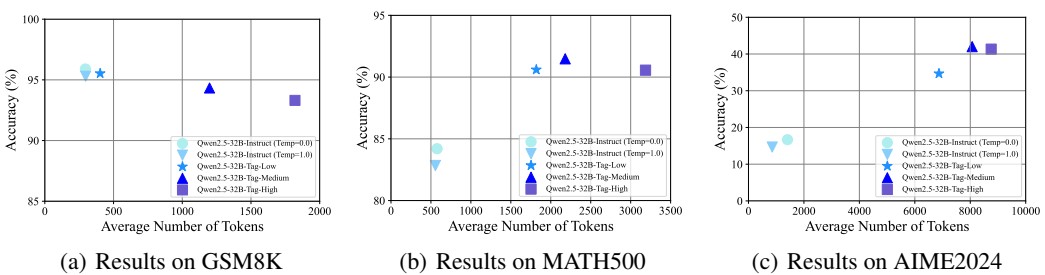

(a) Results on GSM8K  (b) Results on MATH500  (c) Results on AIME2024

Figure 3: The accuracy and the average number of tokens of Qwen2.5-32B-Instruct and Qwen2.5-32B-Tag under different reasoning efforts ("Low", "Medium" and "High") on different benchmarks with varying levels of difficulty.

Preview on all evaluation benchmarks when directly prompted with different prompts in Appendix H for reference. We can draw several interesting conclusions from these results: (1) **A small number o1-like responses is already highly effective in enhancing the reasoning performance of LLMs.** This is also consistent with the previous findings [25, 14]. (2) **Scaling with longer CoTs can bring negative effects to the model's reasoning performance in certain domains**, especially on easy tasks. For example, both LLaMA3.1-8B-Tag and Qwen2.5-32B-Tag perform worse under high reasoning effort compared to the other two reasoning efforts, while consuming significantly more tokens, particularly on GSM8K and MATH500. (3) **There exists an optimal reasoning efforts that varies across different tasks of varying difficulty levels.** As we can see, low reasoning effort consistently works best on GSM8K, while medium and high reasoning efforts are more beneficial for harder question. Furthermore, we display the breakdown results of each model on MATH500 categorized by different problem levels in Appendix F, and the results also show the consistent conclusions on the negative effect of excessively scaling CoT lengths. We also conduct additional experiments on general reasoning tasks beyond mathematical reasoning to demonstrate that the above findings hold true in other tasks as well. The detailed results and discussions are in Appendix J.

To further investigate the impact of scaling with varying CoT length distributions, we calculate the distributions of answers of reasoning effort-conditioned models under different reasoning efforts. Specifically, we calculate the average number of distinct answers in 5 samples per prompt under each reasoning effort. The results are in Table 2. We also display the accuracy on each benchmark for reference. We find a very interesting phenomenon. **Generally, the reasoning effort that achieves the best performance on a certain benchmark also leads to the lowest average number of distinct answers per prompt.** This indicates that, **under the optimal thinking effort, the model can generate the most consistent answers across multiple samplings without either underthinking or overthinking**.

### 3.3 Analysis on the adverse effects of excessive length scaling

Here, we take a deeper step to explore why training on longer CoTs leads to a decline in model's reasoning performance. We randomly selected 100 problems from the tag model's training set along

Table 2: The distribution of distinct answers under different reasoning efforts along with the average accuracy. We highlight the best accuracy and the lowest average number of distinct answers per prompt.

| Model | GSM8K | | MATH500 | | AIME2024 | |
|---|---|---|---|---|---|---|
| | Accuracy | #Answers | Accuracy | #Answers | Accuracy | #Answers |
| *LLaMA3.1 models* | | | | | | |
| LLaMA3.1-8B-Tag-Low | **87.26** | **1.37** | 59.00 | **2.34** | 7.33 | 4.27 |
| LLaMA3.1-8B-Tag-Medium | 87.06 | 1.42 | **61.12** | 2.54 | 7.33 | 4.27 |
| LLaMA3.1-8B-Tag-High | 86.89 | 1.47 | 59.36 | 2.59 | **10.00** | **4.00** |
| *Qwen2.5 models* | | | | | | |
| Qwen2.5-32B-Tag-Low | **95.53** | **1.33** | 90.60 | 1.39 | 34.67 | 3.20 |
| Qwen2.5-32B-Tag-Medium | 94.33 | 1.14 | **91.48** | **1.36** | **42.00** | **3.13** |
| Qwen2.5-32B-Tag-High | 93.31 | 1.15 | 90.56 | 1.40 | 41.33 | 3.30 |

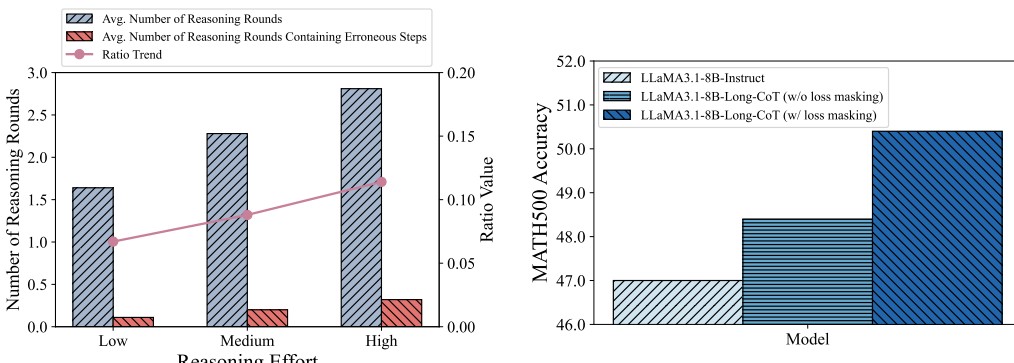

Figure 4: The statistics of responses under different reasoning efforts for training the tag models.

Figure 5: Empirical results of loss masking on erroneous steps. Evaluation temperature is 0.0.

with their responses under three different reasoning efforts. We first follow the existing study [3] to use `gpt-4o` to determine the total number of reasoning rounds contained in each response. Each reasoning round is defined as a complete reasoning process or verification process that contains a final answer. Besides, we further use `gpt-4o` to determine the number of reasoning rounds that contain erroneous steps or wrong final answers. The utilized prompt is shown in Appendix D. We visualize the average number of reasoning rounds, the average number of erroneous reasoning rounds, and the average ratio of erroneous reasoning rounds, on each problem under each reasoning effort in Figure 4. Note that all evaluated responses have correct final answers.

First, we can see that the number of reasoning rounds consistently increases from low reasoning effort to high reasoning effort. It could lead to the overthinking issue in reasoning models [3], where redundant tokens are used to answer the question through repetitive verification. This, however, should not affect the accuracy if all reasoning rounds are correct. Unfortunately, we observe a pattern that **the number and proportion of erroneous reasoning rounds also increase when the reasoning effort becomes higher**. Training the model on more wrong steps would bring adverse effects to the model's reasoning abilities, which can explain why scaling with high reasoning effort leads to worse results. Therefore, we can conclude that while including a certain incorrect and reflective steps can help the model learn to correct errors during inference, an excess of erroneous steps can have a detrimental impact on model's learning.

To further validate the above claim, we conduct controlled validation experiments about training the model on erroneous reasoning steps with subsequent error corrections. Specifically, we first construct long CoTs with reflection and correction from scratch. We obtain initial CoTs with incorrect answers generated by LLaMA3.1-8B-Instruct on MATH training prompts, and use Qwen2.5-72B-Instruct to critique them (i.e., identify the erroneous step and generate suggestions for correction). Then, we enable LLaMA3.1-8B-Instruct to generate self-reflections and corrections based on the critiques, and

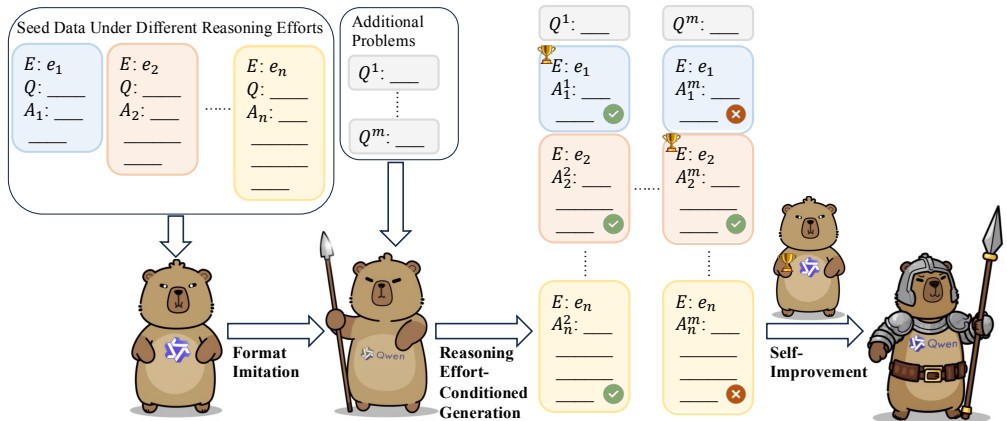

Figure 6: The illustration of our Thinking-Optimal Scaling method. Our method includes three stages: **Format Imitation** enables the base model to learn how to adopt different levels of reasoning effort $e_i$ to perform System-2 thinking, using a small set of seed data. **Reasoning Effort-Conditioned Generation** requires the model to apply System-2 thinking to a large set of problems under different reasoning efforts. **Self-Improvement** select the shortest correct response for each problem among all responses to fine-tune the base model to achieve thinking-optimal test-time scaling.

merge these with the initial responses to form long CoTs. We fine-tune LLaMA3.1-8B-Instruct on the created long CoTs. In the results shown in Figure 5, we can observe that **if we apply loss masking to the tokens in the identified wrong steps (i.e., not calculating loss on these erroneous steps), the performance is better compared to calculating loss on all steps/tokens**. This further validates our claim that training on erroneous steps can negatively impact model performance. Additional experiments and empirical analysis are in Appendix I.

## 4 Thinking-optimal test-time scaling

The above analysis reveals that excessively increasing the response lengths of LLMs can result in negative consequences. This motivates us that an optimal approach to achieve test-time scaling is allowing the model to determine by itself the number of tokens needed to solve each problem. Specifically, for a simple question, if the model can provide a correct answer within a certain number of tokens, further extending the CoTs becomes suboptimal, as it may introduce unnecessary overthinking or even additional erroneous steps into the reasoning process. Conversely, the model should be encouraged to use more tokens for difficult problems if additional reasoning effort can help it to obtain a more reliable and accurate answer.

Thus, we propose a **Thinking-OPtimal Scaling** (**TOPS**) strategy aiming to achieve more effective and efficient test-time scaling for LLM reasoning. We define **a System-2 thinking-optimal response as the shortest correct response that the model can generate using System-2 thinking**: fewer tokens may lead to wrong answer while more tokens causes overthinking. Then, our method includes three stages: Format Imitation, Reasoning Effort-Conditioned Generation, and Self-Improvement. The illustration of our method is shown in Figure 6.

**Format Imitation** First, we require a small set of o1-like responses for a cold start, enabling the model to learn the format of System-2 thinking patterns, including searching, reflecting, verification, backtracking, etc. Preliminary results in Section 3.2 and previous studies [25, 14] have shown that a small number of o1-like responses are sufficient for the model to effectively learn the format of System-2 reasoning. Such a small set of seed data can be manually human-written or generated by existing o1-like models. However, different from previous studies [25, 26] that use a fixed length distribution of seed samples (i.e., directly generated by existing o1-like models), which may not be a thinking-optimal distribution for our base model, we instead create the seed data containing responses under different reasoning efforts (i.e., different length distributions). Specifically, we define a small number of seed problems as $\mathcal{P}_s$, and our goal is to curate a seed dataset $\mathcal{D}_s = \mathcal{D}_{s_1} \cup \cdots \cup \mathcal{D}_{s_n}$, where $\mathcal{D}_{s_i} = \{(e_i, x, y_{e_i}) | x \sim \mathcal{P}_s\}$ represents the responses to seed problems under a specific

reasoning effort $e_i$. This follows the same procedure as data curation for the tag model in Section 3.2. Then, we train the base model on this dataset to obtain the tag model that can apply different levels of reasoning effort to perform System-2 thinking on a given problem:

$$\boldsymbol{\theta}_{tag} = \arg\max_{\boldsymbol{\theta}} \mathbb{E}_{(e_i,x,y_{e_i})\sim\mathcal{D}_s}[P(y_{e_i}|e_i,x,\boldsymbol{\theta})]. \tag{1}$$

**Reasoning Effort-Conditioned Generation** We then use the tag model to generate the solutions on a large number of additional math problems $\mathcal{P}_a$ under different reasoning efforts $e_i$:

$$y_{e_i} \sim \pi(\cdot|e_i,x;\boldsymbol{\theta}_{tag}), \quad x \sim \mathcal{P}_a, \tag{2}$$

where $\pi(\cdot|\boldsymbol{\theta}_{tag})$ denotes the output distribution of the tag model. We select the shortest correct solution $y_{sc}$ among all generations $\{y_{e_1},\cdots,y_{e_n}\}$ as the thinking-optimal response for problem $x$, and obtain a thinking-optimal self-improvement dataset $\mathcal{D}_{TOP} = \{(x,y_{sc})|x\sim\mathcal{P}_a\}$.

**Self-Improvement** After obtaining the thinking-optimal dataset determined by the model itself, we can use it to train the base model, enabling the base model to achieve better self-improvement on System-2 thinking. Specifically, we perform Supervised Fine-Tuning (SFT) to the base model on $\mathcal{D}_{TOP}$:

$$\boldsymbol{\theta}_{TOP} = \arg\max_{\boldsymbol{\theta}} \mathbb{E}_{(x,y_{sc})\sim\mathcal{D}_{TOP}}[P(y_{sc}|x,\boldsymbol{\theta})]. \tag{3}$$

# 5 Experiments and analysis

## 5.1 Experimental settings

**Base Model** We mainly display and compare the results of performing Thinking-Optimal Scaling on Qwen2.5-32B-Instruct, as it serves as an appropriate base model for exploration on test-time scaling according to previous works [30, 25, 26]. To validate the generalizability of our approach, we also perform additional experiments on LLaMA3.1-8B-Instruct, and put the results in Section 5.4.

**Datasets** First, we directly use the model Qwen2.5-32B-Tag created in Section 3.2 as the tag model for reasoning effort-conditioned generation. As mentioned before, this tag model is trained on the seed data that contains 1.3K problems from a subset of NuminaMath, and totally 3.9K responses under three types of reasoning efforts generated by QwQ-32B-Preview. We then use this tag model to generate responses on an additional subset of NuminaMath containing extra 50K problems under different reasoning efforts. On each problem, we sample only 1 response for each reasoning effort, though we believe that performing multiple samplings could further enhance effectiveness. For each problem, we select the shortest correct response among all three responses to the problem as the thinking-optimal response. Finally, we incorporate the responses corresponding to low reasoning effort from the seed data into the above generated dataset, resulting in a thinking-optimal dataset of about 26K samples for self-improvement. We denote the self-improved model created by our method as **Qwen2.5-32B-TOPS**. We then evaluate the performance of our model on three typical math reasoning benchmarks: GSM8K, MATH500, and AIME2024. The additional results on general reasoning tasks can be found in Appendix J.

**Training and Evaluation Details** In the SFT stage, the learning rate is $1\times10^{-5}$, the batch size is 96, and the number of epochs is 2. In inference, the decoding temperature is 1.0, the maximum generation length is 16,384. We report the average accuracy across 5 random seeds in each experiment. The standard deviation results are in Appendix K. In the evaluation of MATH500, we also use `gpt-4o` to assist in identifying missed cases caused by format issues [8].

**Baselines** Besides the base model Qwen2.5-32B-Instruct, we compare our self-improved model with several existing o1-like models that are based on the same base model: (1) **QwQ-32B-Preview**: One of the most popular o1-like reasoning models developed by Qwen Team. (2) **STILL-2-32B** [25]: A System-2 thinking model trained on 3.9K challenging math examples generated by QwQ-32B-Preview and DeepSeek-R1-Lite[7]. (3) **Sky-T1-32B-Preview** [26]: The reasoning model trained on 17K examples generated by QwQ-32B-Preview, including 10K math examples. (4) **Qwen2.5-32B-Random**: After the reasoning effort-based generation, we randomly select a correct solution on each problem rather than the shortest one to form a thinking-suboptimal dataset, and train the base model on this dataset.

---

[7]`https://api-docs.deepseek.com/news/news1120`

Table 3: The results of our self-improved (Qwen2.5-32B-TOPS) and further iteratively self-improved models (Qwen2.5-32B-TOPS-Iter) compared to existing o1-like models using the same base model on GSM8K, MATH500, and AIME2024. In each setting, the underlined value represents the best result for System-1 thinking models, while the bold **value** indicates the best result for System-2 thinking models.

| Model | GSM8K | | MATH500 | | AIME2024 | |
|---|---|---|---|---|---|---|
| | Accuracy | #Tokens | Accuracy | #Tokens | Accuracy | #Tokens |
| *System-1 thinking models* | | | | | | |
| Qwen2.5-32B-Instruct (Temp. $= 0.0$) | 95.91 | 295.01 | 84.20 | 576.89 | 16.67 | 1407.43 |
| Qwen2.5-32B-Instruct (Temp. $= 1.0$) | 95.30 | 296.98 | 82.84 | 555.65 | 14.67 | 855.62 |
| *System-2 thinking models* | | | | | | |
| QwQ-32B-Preview | 95.23 | 761.01 | **92.02** | 2416.23 | 45.33 | 7636.63 |
| STILL-2-32B | 95.47 | 570.64 | 91.40 | 2005.28 | 45.33 | 6656.11 |
| Sky-T1-32B-Preview | 94.82 | 695.66 | 89.48 | 2022.07 | 35.33 | 5351.29 |
| Qwen2.5-32B-Random | 95.00 | 938.45 | 90.16 | 2670.19 | 39.33 | 7691.30 |
| Qwen2.5-32B-TOPS (ours) | **95.82** | 412.24 | 91.48 | 1883.29 | 43.33 | 7260.26 |
| Qwen2.5-32B-TOPS-Iter-SFT (ours) | 95.45 | 366.14 | 90.76 | 1701.11 | 44.00 | 6611.89 |
| Qwen2.5-32B-TOPS-Iter-DPO (ours) | 95.80 | 384.81 | 91.60 | 1731.72 | **46.00** | 6426.62 |

## 5.2 Main results

The results of each model are displayed in Table 3. Besides the accuracy, we also report the number of CoT tokens used by each model on each dataset.[8]

First, we can see that the model trained under thinking-optimal samples (Qwen2.5-32B-TOPS) consistently performs better than the model trained under thinking-suboptimal samples (Qwen2.5-32B-Random). This helps to revalidate our motivation that scaling with shortest correct responses, as determined by the base model itself using System-2 thinking, is the most effective approach to achieve optimal test-time scaling. Second, compared to distillation-based models STILL-2-32B and Sky-T1-32B-Preview, our self-improvement-based model Qwen2.5-32B-TOPS achieves better results across the board, except for AIME2024, where it slightly underperforms STILL-2-32B. However, note that STILL-2-32B uses a greater number of high-quality distilled samples (3.9K) including more challenging problems from AIME1983-2023, whereas our model achieves comparable performance using only 1.3K seed samples and effective self-improvement strategy.

Regarding the reasoning efforts (i.e., the number of reasoning tokens) used by each model to solve different difficulty levels of tasks, we observe that Qwen2.5-32B-TOPS uses fewer tokens on easier tasks like GSM8K compared to other models, effectively mitigating the issue of overthinking [3]. On the other hand, it tends to spend more time thinking on harder problems such as AIME2024. The comparison of reasoning tokens used by different models across various domains reflects our model's ability to exhibit adaptive reasoning depths.

## 5.3 Results of iterative self-improvement

To further enhance the reasoning performance of our model on challenging problems, we perform iterative self-improvement on Qwen2.5-32B-TOPS. Specifically, we select additional 4500 MATH problems [13] (which have not appeared in the previously used problems) and the problems from AIME1983-2023. On each problem, we sample 8 responses from Qwen2.5-32B-TOPS. Then, we select the shortest correct response among 8 responses as the chosen response. One iterative self-improvement approach is to further supervised fine-tune Qwen2.5-32B-TOPS on the dataset composed of all chosen responses (shortest correct responses), resulting in **Qwen2.5-32B-TOPS-Iter-SFT**. Besides, we can also perform preference optimization. Specifically, if there are responses with incorrect final answers, we select the longest incorrect response as the rejected response to improve reasoning capability. Additionally, we include preference pairs where the rejected response is the shortest wrong response if there exists a wrong response that is shorter than the shortest correct

---

[8]For STILL-2-32B and Sky-T1-32B-Preview, we only calculate the number of tokens in the thought part and do not include the tokens in the summary part.

Table 4: The self-improvement results on LLaMA3.1-8B-Instruct. In each setting, the underlined value represents the best result for System-1 thinking models, while the bold **value** indicates the best result for System-2 thinking models.

| Model | GSM8K | | MATH500 | | AIME2024 | |
|---|---|---|---|---|---|---|
| | Accuracy | #Tokens | Accuracy | #Tokens | Accuracy | #Tokens |
| *System-1 thinking models* | | | | | | |
| LLaMA3.1-8B-Instruct (Temp. = 0.0) | 82.18 | 262.23 | 47.00 | 1801.76 | 6.67 | 5506.30 |
| LLaMA3.1-8B-Instruct (Temp. = 1.0) | 76.21 | 233.08 | 39.60 | 733.56 | 4.67 | 1691.88 |
| *System-2 thinking models* | | | | | | |
| LLaMA3.1-8B-Random-SFT | 87.94 | 1051.05 | 60.52 | 3627.23 | 4.67 | 8165.69 |
| LLaMA3.1-8B-TOPS-SFT | **88.54** | 571.10 | **61.28** | 3254.01 | **8.00** | 7392.59 |

response, to avoid causing the model to underthink. After obtaining the preference dataset, we perform Direct Preference Optimization (DPO) [32] on Qwen2.5-32B-TOPS to get **Qwen2.5-32B-TOPS-Iter-DPO**. Detailed experimental settings are in Appendix E.

The results of iterative self-improvement are in Table 3. As we can observe, further SFT is mainly effective in shortening the CoT lengths but does not necessarily improve reasoning performance. Preference optimization improves both the efficiency and the effectiveness, resulting in a reasoning model that is comparable to QwQ-32B-Preview.

## 5.4 Results on LLaMA3.1-8B-Instruct

Here, we display the results of performing our Thinking-Optimal Test-Time Scaling strategy on LLaMA3.1-8B-Instruct in Table 4. The experimental setups are consistent with that in the main experiments above. The results demonstrate the generalizability of our method on other model architectures.

## 6  Conclusion

In this work, we aim to explore a potential issue under current pursuit of test-time scaling. Through empirical analysis on mathematical reasoning tasks, we first demonstrate that overly long CoTs can negatively impact the model's reasoning performance in certain domains, emphasizing the need for optimal CoT length scaling. To tackle this, we propose a Thinking-Optimal Scaling strategy, which first leverages a small set of seed data to teach LLMs to adopt varying levels of reasoning effort to perform System-2 thinking. Then, our approach allows models to identify the shortest correct response for self-improvement, leading to more efficient and effective System-2 thinking. Experimental results show that our self-improved and iteratively self-improved models, based on Qwen2.5-32B-Instruct, outperform existing distillation-based o1-like models and achieve comparable performance with QwQ-32B-Preview across various math benchmarks.

## 7  Limitations

There are some limitations in the current work: (1) Our analysis mainly focuses on the math reasoning domain, because math tasks offer relatively accurate and reliable performance verification. However, we believe investigating the potential impact of long CoTs in other domains is important, which can be a promising direction in future work. We also provide some preliminary results in the general reasoning domain in Appendix J. (2) This work primarily focuses on the SFT setting. While investigating effective SFT and distillation is very important and practical—as highlighted in concurrent studies [25, 38]—exploring the impact of CoT length in the reinforcement learning (RL) setting is another compelling direction. We believe that our findings can be extended to RL-based scaling. Specifically, since RL assigns positive rewards (e.g., 1.0) to all solutions with correct final answers, shorter correct solutions with fewer erroneous reasoning steps should be preferred over longer correct ones with more errors. Over-rewarding the latter may encourage the model to generate incorrect intermediate steps first and rely on its imperfect self-correction ability to correct erroneous steps, rather than producing the correct answer in one go.

## Acknowledgements

We sincerely thank all the anonymous reviewers and (S)ACs for their valuable comments and thoughtful suggestions. This work was supported by The National Natural Science Foundation of China (No. 62376273 and No.U2436209) and Beijing Natural Science Foundation (L253001).

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

Table 5: CoT token count returned from o1-mini API and estimated by Qwen2.5 tokenizer.

| Method | MATH500 | AIME2024 |
|---|---|---|
| o1-mini API | 1122.07 | 4861.87 |
| Qwen2.5 Tokenizer | 1110.88 | 4972.08 |

## A Impact statement

This work aims to explore the effectiveness and limitations of o1-like test-time scaling. Our goal is to enhance both the efficiency and effectiveness of test-time scaling in a more thinking-optimal way. These findings highlight the importance of adaptive reasoning efforts and provide a promising direction for more effectively enhancing LLM reasoning capabilities.

## B Details on estimating the number of tokens in hidden CoT of o1-mini by Qwen2.5 tokenizer

In the preliminary analysis in Section 3.1, we use Qwen2.5 tokenizer to calculate the number of tokens in the CoTs generated by each model for a fair comparison. However, o1-mini does not expose the internal CoTs to users, but only shows the summary parts $S$, along with the number of reasoning tokens $n_r^{o1}$ and total number completion tokens $n_c^{o1}$ (the sum of reasoning tokens and summary tokens) measured by o1-mini tokenizer. Therefore, we choose to estimate the number of tokens in its hidden CoTs measured by the Qwen2.5 tokenizer using $S$, $n_r^{o1}$ and $n_c^{o1}$. Specifically, we denote the number of tokens of the summary part measured by Qwen2.5 tokenizer as $n_s^{qwen}$, then the estimated number of tokens of hidden CoT by Qwen2.5 tokenizer can be calculated as $n_s^{qwen} \times \frac{n_r^{o1}}{n_c^{o1} - n_r^{o1}}$.

We also display the comparison results on the CoT token count returned by the o1-mini API and the number of tokens we estimated using the Qwen2.5 tokenizer in Table 5. As we can see, the numbers do not differ significantly. Thus, we use the estimation results from the Qwen2.5 tokenizer in the main text in order to make a fair comparison of the number of reasoning tokens used by different models.

## C Prompts for generating reasoning responses under different reasoning efforts

We put the system prompts for QwQ-32B-Preview to generate responses under different reasoning efforts in Figure 7.

## D User prompt for `gpt-4o` to determine the number of (erroneous) reasoning rounds

We display the user prompt for `gpt-4o` to help determine the number of total and erroneous reasoning rounds in Figure 8.

## E Experimental settings

### E.1 Training settings for tag models

When creating two tag models, the learning rate is $1 \times 10^{-5}$, the batch size is 32. The number of epochs is 3 for Qwen2.5-32B-Tag and 5 for LLaMA3.1-8B-Tag. The training is performed on 8×NVIDIA H100 80G.

**Low Reasoning Effort:** You have extremely limited time to think and respond to the user's query. Every additional second of processing and reasoning incurs a significant resource cost, which could affect efficiency and effectiveness. Your task is to prioritize speed without sacrificing essential clarity or accuracy. Provide the most direct and concise answer possible. Avoid unnecessary steps, reflections, verification, or refinements UNLESS ABSOLUTELY NECESSARY. Your primary goal is to deliver a quick, clear and correct response.

**Medium Reasoning Effort:** You have sufficient time to think and respond to the user's query, allowing for a more thoughtful and in-depth answer. However, be aware that the longer you take to reason and process, the greater the associated resource costs and potential consequences. While you should not rush, aim to balance the depth of your reasoning with efficiency. Prioritize providing a well-thought-out response, but do not overextend your thinking if the answer can be provided with a reasonable level of analysis. Use your reasoning time wisely, focusing on what is essential for delivering an accurate response without unnecessary delays and overthinking.

**High Reasoning Effort:** You have unlimited time to think and respond to the user's question. There is no need to worry about reasoning time or associated costs. Your only goal is to arrive at a reliable, correct final answer. Feel free to explore the problem from multiple angles, and try various methods in your reasoning. This includes reflecting on reasoning by trying different approaches, verifying steps from different aspects, and rethinking your conclusions as needed. You are encouraged to take the time to analyze the problem thoroughly, reflect on your reasoning promptly and test all possible solutions. Only after a deep, comprehensive thought process should you provide the final answer, ensuring it is correct and well-supported by your reasoning.

Figure 7: System prompts for QwQ-32B-Preview with varying levels of reasoning effort.

You will be provided with a math problem and a solution generated by a reasoning model.
The model's response may consist of multiple reasoning rounds.
One reasoning round is a part of the full model generation and is defined as a complete reasoning process or verification process that explicitly contains the final answer.
Your task is to carefully analyze the response to determine the number of reasoning rounds it contains, and identify how many of these solutions contain erroneous steps, including intermediate erroneous steps or erroneous final answer that is different from the ground truth answer.
After you reasoning process, please give your final conclusions as "#### Number of rounds: <number>" and "#### Number of wrong rounds: <number>".

Problem: {question}
Solution: {solution}
Ground Truth Answer: {answer}

Figure 8: User prompt for `gpt-4o` to determine the number of reasoning rounds.

### E.2 Training settings in format imitation

In the format imitation stage, we perform SFT on the base model Qwen2.5-32B-Instruct on a small subset of seed data containing 1.3K problems sampled from NuminaMath along with responses with varying lengths for each problem. The statistics of the seed data is shown in Table 1. In SFT stage, the learning rate is $1 \times 10^{-5}$, the batch size is 32, the number of epochs is 3. The training is performed on 8×NVIDIA H100 80G.

### E.3 Training settings in self-improvement

In the self-improvement stage, we perform SFT on Qwen2.5-32B-Instruct on the curated thinking-optimal dataset for 2 epochs. The learning rate is $1 \times 10^{-5}$, and the batch size is 96. The training is performed on 4×NVIDIA H100 80G.

### E.4 Training settings in iterative self-improvement

In the iterative self-improvement stage, for Qwen2.5-32B-TOPS-Iter-SFT, the learning rate is $1 \times 10^{-6}$, the batch size is 32, and we set the training epoch to 1. For Qwen2.5-32B-TOPS-Iter-DPO, the learning rate is $5 \times 10^{-7}$, the batch size is 32, the training epoch is 3. The training is performed on 8×NVIDIA H100 80G.

### E.5 Evaluation settings

For all o1-like models, we set the decoding temperature to 1.0 and average the results over 5 random seeds for each evaluation experiment. The maximum generation length is 16,384. All evaluations are conducted on $4 \times$ NVIDIA A100 80G.

Table 6: Breakdown results of tag models on MATH500 categorized by problem difficulty levels.

| Model | Level-1 | | Level-2 | | Level-3 | | Level-4 | | Level-5 | |
|---|---|---|---|---|---|---|---|---|---|---|
| | Acc. | #Tokens | Acc. | #Tokens | Acc. | #Tokens | Acc. | #Tokens | Acc. | #Tokens |
| LLaMA3.1-8B-Tag-Low | **92.09** | 1415.00 | 78.89 | 1702.68 | 68.38 | 2238.64 | 54.22 | 3386.63 | 32.24 | 4719.16 |
| LLaMA3.1-8B-Tag-Medium | 87.44 | 2123.17 | **79.11** | 2775.34 | **72.19** | 3541.26 | **57.81** | 4467.70 | **35.07** | 6267.16 |
| LLaMA3.1-8B-Tag-High | 87.44 | 2924.74 | 78.67 | 3561.78 | 71.62 | 4447.92 | 54.22 | 5242.71 | 32.69 | 6735.32 |
| Qwen2.5-32B-Tag-Low | **97.67** | 673.82 | 95.33 | 987.61 | **97.14** | 1098.93 | 90.31 | 1802.25 | 80.30 | 3330.60 |
| Qwen2.5-32B-Tag-Medium | 96.74 | 1757.15 | **96.22** | 1761.37 | **97.14** | 1983.42 | **90.94** | 2692.74 | **82.69** | 4344.62 |
| Qwen2.5-32B-Tag-High | 96.74 | 2077.88 | 95.33 | 2291.11 | 95.43 | 2479.56 | 89.84 | 3223.67 | 82.24 | 4659.30 |

Table 7: Raw length distributions (response length rankings) of different reasoning-effort prompted responses on QwQ-32B-Preview.

| Effort | Longest % | Middle % | Shortest % |
|---|---|---|---|
| Low | 15.2 | 18.3 | 66.5 |
| Medium | 36.4 | 45.4 | 18.2 |
| Hight | 48.4 | 36.2 | 15.4 |

## F   Breakdown results of tag models on MATH500

We display the breakdown results of tag models on MATH500 under different difficulty levels in Table 6. The results also validate the claim that **continuously increasing the reasoning effort can indeed bring adverse effects**, especially on lower-level problems (e.g., 92.09 (Low) -> 87.44 (Medium/High) and 97.67 (Low) -> 96.74 (Medium/High) on Level 1 problems for LLaMA3.1-8B-Tag and Qwen2.5-32B-Tag separately).

## G   Response length distribution of QwQ-32B-Preview under different reasoning effort-based system prompts

We put the initial response length distribution of different reasoning-effort prompted responses on QwQ-32B-Preview in Table 7. As we can see, in nearly 15% of cases, the low reasoning-effort prompted responses are even the longest among all three responses. This indicates that only using direct prompting is unreliable to explore the impact of CoT length on reasoning performance. Therefore, when training tag models, for each training problem, we reorder the responses to ensure that the response lengths under different reasoning efforts follow an consistently increasing order from low to medium to high. In such a controlled setting, we can fairly observe the impact of CoT length on reasoning performance.

## H   Performance of QwQ-32B-Preview and QwQ-32B when directly prompted with different reasoning effort-based prompts

Here, we display the performance of QwQ-32B-Preview [30] and QwQ-32B [31] on all three evaluation benchmarks when directly prompted with different reasoning effort-based prompts. Since QwQ-32B includes both reasoning and summary components in its responses and typically generates much longer CoTs, we extended its `max_seq_len` to 32K to allow for sufficient reasoning and summarization. The full results are in Table 8. These results re-validate our main claim: **longer CoTs may not necessarily lead to better performance**.

Table 8: The performance of QwQ-32B-Preview and QwQ-32B on three benchmarks when directly prompting the model with various reasoning effort-based prompts.

| Reasoning Effort | GSM8K | | MATH500 | | AIME2024 | |
|---|---|---|---|---|---|---|
| | Accuracy | #Tokens | Accuracy | #Tokens | Accuracy | #Tokens |
| *QwQ-32B-Preview* | | | | | | |
| Low | **93.95** | 418.26 | **90.24** | 1592.29 | 42.00 | 5152.97 |
| Medium | 93.56 | 844.32 | 89.16 | 2356.29 | **44.00** | 6413.97 |
| High | 92.78 | 1112.97 | 88.36 | 2684.87 | 41.33 | 6678.50 |
| *QwQ-32B* | | | | | | |
| Low | **96.56** | 568.49 | 95.76 | 2532.82 | 74.00 | 11390.70 |
| Medium | 96.50 | 1152.86 | **96.08** | 3708.79 | 78.00 | 12969.53 |
| High | 96.36 | 1752.29 | 95.80 | 4198.67 | **80.67** | 13194.29 |

Table 9: The performance of the model trained on data with high reasoning effort after filtering out all solutions that are identified to contain erroneous steps.

| Model | GSM8K | | MATH500 | | AIME2024 | |
|---|---|---|---|---|---|---|
| | Accuracy | #Tokens | Accuracy | #Tokens | Accuracy | #Tokens |
| Qwen2.5-32B-Tag-High | 93.31 | 1820.17 | **90.56** | 3185.75 | **41.33** | 8753.87 |
| Qwen2.5-32B-Tag-High-Filtered | **94.87** | 1478.10 | 90.00 | 2783.98 | 36.33 | 8049.29 |

# I Results of fine-tuning on samples after filtering out solutions with erroneous steps

We conduct additional experiments on supporting the claim that including more erroneous rounds in the training stage can lead to greater adverse effects. Specifically, we filter out those solutions that contain erroneous steps (identified by GPT-4.1) under the high reasoning effort, and fine-tune Qwen2.5-32B-Instruct on the filtered data, which yields Qwen2.5-32B-Tag-High-Filtered. The results are in Table 9.

We have some interesting findings: (1) After removing samples containing erroneous rounds, the trained model produces much shorter CoTs. The reason is that the solutions removed are usually very lengthy as they contain more self-reflections and self-corrections, thus the average length of the dataset after removing these samples is much shorter. (2) Removing samples containing erroneous rounds brings significant performance improvement on GSM8K, while causing certain performance degradation on harder benchmarks MATH500 and AIME2024. The reason is that after removing those samples, the model cannot learn to effectively perform deeper thinking such as correcting the errors it could make in previous rounds. GSM8K is relatively simple, so the model does not need to engage in excessive reflection and correction. Therefore, removing these behaviors actually improves model performance. On more challenging datasets, the advantages of longer reasoning chains that include reflections and corrections become apparent. Moreover, by comparing with the results in Figure 5 in the main text, we find that instead of directly removing correct solutions that contain erroneous steps, a more effective approach is to apply loss masking specifically to the wrong steps. The latter strategy allows the model to retain the ability to learn how to correct previous mistakes, without explicitly learning from the erroneous steps themselves.

# J Results on general reasoning tasks

Here, we additionally prompt QwQ-32B-Preview to generate responses under different reasoning effort-based prompts on a subset of the WebInstruct-verified dataset [23], and curate a seed general reasoning dataset that contains correct responses with varying reasoning efforts. We then fine-tune Qwen2.5-7B-Instruct [29] on the seed dataset to get Qwen2.5-7B-Tag-General, and evaluate

Table 10: The performance of Qwen2.5-7B-based models on MMLU-Pro and GPQA-Diamond

| Model | MMLU-Pro | | GPQA-Diamond | |
|---|---|---|---|---|
| | Accuracy | #Tokens | Accuracy | #Tokens |
| ***System-1 thinking models*** | | | | |
| Qwen2.5-7B-Instruct (Temp. $= 0.0$) | 52.46 | 401.38 | 34.85 | 592.73 |
| Qwen2.5-7B-Instruct (Temp. $= 1.0$) | 51.49 | 379.84 | 33.84 | 537.41 |
| ***Tag models*** | | | | |
| Qwen2.5-7B-Tag-General-Low | 56.00 | 1674.60 | 31.82 | 2808.88 |
| Qwen2.5-7B-Tag-General-Medium | 55.92 | 2341.27 | 36.87 | 3931.13 |
| Qwen2.5-7B-Tag-General-High | 55.81 | 2632.05 | 32.83 | 4238.11 |
| ***System-2 thinking models*** | | | | |
| Qwen2.5-7B-Random-General | 55.86 | 1960.87 | 34.34 | 3385.70 |
| Qwen2.5-7B-TOPS-General (ours) | **56.50** | 1788.74 | **38.38** | 3083.76 |

Table 11: The detailed standard deviation results on LLaMA3.1-8B-Instruct.

| Model | GSM8K | MATH500 | AIME2024 |
|---|---|---|---|
| ***System-1 thinking models*** | | | |
| LLaMA3.1-8B-Instruct (Temp. $= 0.0$) | 82.18 ($\pm$ 0.00) | 47.00 ($\pm$ 0.00) | 6.67 ($\pm$ 0.00) |
| LLaMA3.1-8B-Instruct (Temp. $= 1.0$) | 76.21 ($\pm$ 0.66) | 39.60 ($\pm$ 0.84) | 4.67 ($\pm$ 2.98) |
| ***System-2 thinking models*** | | | |
| LLaMA3.1-8B-Random-SFT | 87.94 ($\pm$ 0.33) | 60.52 ($\pm$ 1.38) | 4.67 ($\pm$ 1.83) |
| LLaMA3.1-8B-TOPS-SFT | **88.54** ($\pm$ 0.26) | **61.28** ($\pm$ 0.73) | **8.00** ($\pm$ 1.83) |

its performance on MMLU-Pro [41] and GPQA-Diamond [33] using the same reasoning effort-conditioned prompting strategy as in Section 3.2. We only sample once for each prompt considering the evaluation cost. The results are displayed in Table 10. **The findings are consistent with the results in the math reasoning domain that excessive scaling with longer CoTs can bring negative effects to the model's performance in general reasoning tasks.** Also, the model needs more reasoning effort to perform well on the more challenging task GPQA-Diamond.

Following the above setup, we then perform our thinking-optimal scaling on a held-out set of the WebInstruct-verified dataset to get Qwen2.5-7B-TOPS-General and perform random scaling to get Qwen2.5-7B-Random-General. The evaluation results in Table 10 show that **our method can also work well for general reasoning tasks**.

# K   Standard deviation results

We put the detailed standard deviation results in Table 11 and Table 12 for reference.

Table 12: The detailed standard deviation results on Qwen2.5-32B-Instruct.

| Model | GSM8K | MATH500 | AIME2024 |
|---|---|---|---|
| ***System-1 thinking models*** | | | |
| Qwen2.5-32B-Instruct (Temp. $= 0.0$) | 95.91 ($\pm$ 0.00) | 84.20 ($\pm$ 0.00) | 16.67 ($\pm$ 0.00) |
| Qwen2.5-32B-Instruct (Temp. $= 1.0$) | 95.30 ($\pm$ 0.36) | 82.84 ($\pm$ 0.65) | 14.67 ($\pm$ 5.06) |
| ***System-2 thinking models*** | | | |
| QwQ-32B-Preview | 95.23 ($\pm$ 0.39) | **92.02** ($\pm$ 0.87) | 45.33 ($\pm$ 3.80) |
| STILL-2-32B | 95.47 ($\pm$ 0.26) | 91.40 ($\pm$ 0.80) | 45.33 ($\pm$ 6.06) |
| Sky-T1-32B-Preview | 94.82 ($\pm$ 0.33) | 89.48 ($\pm$ 0.78) | 35.33 ($\pm$ 2.98) |
| Qwen2.5-32B-Random | 95.00 ($\pm$ 0.31) | 90.16 ($\pm$ 0.95) | 39.33 ($\pm$ 2.79) |
| Qwen2.5-32B-TOPS (ours) | **95.82** ($\pm$ 0.15) | 91.48 ($\pm$ 0.59) | 43.33 ($\pm$ 2.36) |
| Qwen2.5-32B-TOPS-Iter-SFT (ours) | 95.45 ($\pm$ 0.14) | 90.76 ($\pm$ 0.71) | 44.00 ($\pm$ 3.65) |
| Qwen2.5-32B-TOPS-Iter-DPO (ours) | 95.80 ($\pm$ 0.13) | 91.60 ($\pm$ 0.62) | **46.00** ($\pm$ 3.65) |

