# OpenReview forum: "Towards Thinking-Optimal Scaling of Test-Time Compute for LLM Reasoning"
_NeurIPS.cc/2025/Conference — NeurIPS 2025 poster_

### Official Review · Reviewer_B8YC · 2025-07-01

**Clarity:** 3
**Significance:** 2
**Originality:** 2
**Rating:** 4
**Confidence:** 4

**Summary:**

This paper investigates the impact of test-time compute scaling via Chain of Thoughts (CoT) length on Large Language Model (LLM) reasoning. The authors challenge the common assumption that longer CoTs always improve performance, demonstrating through mathematical reasoning tasks that excessive CoT length can harm accuracy, particularly in easier problems. They identify an optimal CoT length distribution varying by task difficulty and propose the Thinking-Optimal Scaling (TOPS) strategy. TOPS trains a "tag" model on diverse reasoning efforts to generate responses of different lengths, then selects the shortest correct response for self-improvement. Experiments on Qwen2.5-32B-Instruct show that TOPS outperforms distillation-based models on math benchmarks (GSM8K, MATH500, AIME2024) and matches QwQ-32B-Preview's performance. The approach addresses overthinking by adapting reasoning effort to task difficulty, reducing token usage on easy tasks while maintaining effectiveness on hard ones.

**Questions:**

Please see the Weaknesses above.

**Ethical Concerns:**

["NO or VERY MINOR ethics concerns only"]

**Final Justification:**

This paper focuses on an important topic of LLM reasoning and is interesting. The authors have addressed some of my concerns, even though the empirical evidence is not strong. I have read the reviews and responses, including reviews from other reviewers. The quality of the revised paper, based on the rebuttal, can be improved. As a result, I have raised my score.

**Limitations:**

yes

**Quality:**

2

**Strengths And Weaknesses:**

Strengths
- Thoroughly evaluates performance on multiple mathematical reasoning benchmarks, showcasing the method's effectiveness.
- The structure of the paper facilitates easy understanding.


Weaknesses
- TOPS relies on QwQ-32B-Preview for seed data generation. Performance may vary with weaker base models. And there is no analysis of model-agnostic generalization (e.g., applying TOPS to non-Qwen architectures).
- While limitations note RL as future work and can be extended (Section 7), TOPS is only tested in SFT settings. The training dynamics of the RL-based reasoning models can be very different from SFT models.

---

> ### Author Rebuttal · Authors · 2025-07-31
>
> We sincerely thank you for your great efforts on reviewing our paper. We are glad that you think our experiments are thorough and our method is effective. We make the following response to address your questions.
>
> **Q1:** Regarding the performance on weaker base models and the generalization to non-Qwen architectures.
>
> **A1:** Thank you and your question is what we have considered. We have performed our method on LLaMA3.1-8B-Instruct, which is an non-Qwen weaker base model. The results are already put in Table 6 in Appendix I. And we also display in the following Table 1 for your reference. Furthermore, we perform additional experiments on Qwen2.5-7B-Instruct in general reasoning domain (the training data is from WebInstruct-Verified [1], and the evaluation data includes MMLUPro and GPQA-Diamond), and show the self-improvements in the following Table 2. **The results show that thinking-optimal scaling can also consistently improve the performance of weaker base models compared with random scaling in diverse domains, demonstrating the generalizability of our method.**
>
>
> Table 1: The self-improvement results of LLaMA3.1-8B-Instruct.
>
> |Model|GSM8K|MATH500|AIME2024|
> |-|-|-|-|
> |LLaMA3.1-8B-Instruct (Temp=0.0)|82.18|47.00|6.67|
> |LLaMA3.1-8B-Instruct (Temp=1.0)|76.21|39.60|4.67|
> |LLaMA3.1-8B-Random|87.94|60.52|4.67|
> |LLaMA3.1-8B-TOPS (ours)|88.54|61.28|8.00|
>
> Table 2: The self-improvement results of Qwen2.5-7B-Instruct on MMLU-Pro and GPQA-Diamond.
>
> |Model|MMLU-Pro|GPQA-Diamond|
> |-|-|-|
> |Qwen2.5-7B-Instruct (Temp=0.0)|52.46|34.85|
> |Qwen2.5-7B-Instruct (Temp=1.0)|51.49|33.84|
> |Qwen2.5-7B-Random-General|55.86|34.34|
> |Qwen2.5-7B-TOPS-General (ours)|**56.50**|**38.38**|
>
> **Q2:** Regarding future work on explorations in the RL setting.
>
> **A2:** **The primary scope and contribution of our work is to uncover the potential negative effects of scaling with excessively long chains of thought (CoTs).** Through comprehensive empirical analysis, we propose an efficient and effective self-improvement strategy, and demonstrate its effectiveness primarily in the SFT/DPO setting. We agree that **investigating the impact of CoT length in the RL setting is a valuable direction for future research, but it is out of the scope of this work**. As discussed in the Limitations section, our findings are also applicable to RL-based scaling. In particular, since RL typically assigns positive rewards (e.g., 1.0) to all solutions with correct final answers, it is preferable to favor shorter correct solutions with fewer erroneous reasoning steps over longer correct ones that contain more errors. Over-rewarding the latter may inadvertently encourage the model to generate incorrect intermediate steps and rely on its imperfect self-correction ability, rather than producing the correct answer directly.
>
> We hope the above clarifications address your concerns. We are glad to have further discussions if you have any further questions.
>
> [1] Ma, Xueguang, et al. "General-reasoner: Advancing llm reasoning across all domains." arxiv 2025

---

> > ### Author Response · Authors · 2025-08-05
> > **Looking forward to your feedback**
> >
> > Dear Review B8YC,
> >
> > We sincerely thank you again for your helpful reviews and comments on our paper. We have followed your suggestions to conduct additional experiments to demonstrate the generalizability of our observations and proposed method on general reasoning tasks and non-Qwen architectures. As the author-reviewer discussion deadline is approaching, we would like to know if you have any further questions, and we are glad to continue the discussion if there are any.
> >
> > We are sincerely looking forward to your feedback, and your support means a lot to us! Thank you.
> >
> > Best regards,
> >
> > Authors

---

> > > ### Comment · Reviewer_B8YC · 2025-08-05
> > >
> > > Thank you for your reply. I have updated the score.

---

> > > > ### Author Response · Authors · 2025-08-05
> > > > **Thank you for your support**
> > > >
> > > > We sincerely thank you for your positive feedback and your great support! We are gald that our response has addressed your questions. We will incorporate your feedback into the revision.

---

### Official Review · Reviewer_AhwA · 2025-07-03

**Clarity:** 3
**Significance:** 3
**Originality:** 3
**Rating:** 5
**Confidence:** 4

**Summary:**

This paper studies how the length of Chain-of-Thought affect model’s reasoning abilities. The authors found that excessively scaling the Chain-of-Thought length can impair the reasoning performance of LLMs. They further found that there exists some optimal length depending on the difficulty of the problems. Based on those findings, they proposed a Thinking-Optimal Scaling strategy to finetune the system-1 models and demonstrate the effectiveness through experiments.

**Questions:**

I will update the score once either or both of the weakness points I raised in the previous section is addressed. Additionally, the authors study the problem in the supervised finetuning setup, and it remains a question whether the same finding is relevant and can generalize to models trained with RL like o1 and Deepseek r1. This may be interesting to include but I understand if this is outside the scope of the paper.

**Ethical Concerns:**

["NO or VERY MINOR ethics concerns only"]

**Final Justification:**

Overall, this is an interesting paper addressing a timely and relevant topic. The authors have adequately responded to all my concerns and questions. While some arguments provided during the rebuttal are not entirely supported by experimental evidence, the overall quality of the paper has improved. As a result, I have raised my score to 5.

**Limitations:**

The authors have included a limitation section.

**Quality:**

3

**Strengths And Weaknesses:**

Overall, the paper is well-written. The findings that excessively longer CoT may harm the reasoning performance contribute to the understanding of test-time scaling. However, there are a few weaknesses.

1.	One potential issue of the method is the usage of system prompt to control the CoT length. Specifically, the authors admit in line 138-140 that QwQ-32B-Preview model they use to generate the responses has relatively poor instruction following abilities. This questions the validity of using system prompt as a way to control the CoT length. Specifically, the results they found may just be because of the poor alignment. What is the performance of QwQ-32B-Preview on the benchmark when prompting the model with different reasoning efforts? Does prompting them for different reasoning efforts affect their performance? The authors should include this result as well.

2.    The author finds that the number of erroneous reasoning rounds also increase when reasoning effort becomes higher in Fig. 4. However, the ratios of erroneous reasoning rounds make more sense than the absolute number of erroneous reasoning rounds. Although erroneous rounds increase, the correct reasoning rounds also increase. Additional experiments are needed if the author wants to claim that including more erroneous rounds in the training stage can lead to greater adverse effects. For example, the authors can filter out those with erroneous steps and finetune and check if the performance of the model trained on data with high reasoning effort degrade or not. I appreciate that the authors conduct controlled experiments and include the results in Fig. 5. However, the dataset is different. Why not just using the same data presented in Fig. 4 and masking out the loss with erroneous steps to demonstrate the point? The dataset generated for this controlled experiment is different from the SFT data the authors use for all other experiments. This raises concerns for consistency.

---

> ### Author Rebuttal · Authors · 2025-07-31
>
> We sincerely thank you for your positive review and your thoughtful suggestions. We are encouraged that you believe our paper is well-written and that our findings contribute to the understanding of test-time scaling. We make the following response to address your remaining questions.
>
> **Q1:** Regarding the usage of system prompts to control the CoT length.
>
> **A1:** Your concern is exactly what we have considered. **To solve this issue, and as explained in Lines 141-143, we do not use the responses from QwQ-32B-Preview under different prompts directly. Instead, we filter and reorder the correct responses from QwQ-32B-Preview under the three reasoning efforts to ensure that, for the same problem, the lengths of the three corresponding responses increase sequentially and significantly from low to medium to high.** This makes the length distributions of the responses under different reasoning efforts clearly distinguishable. Then, we set different system prompts for different subsets of responses under the three reasoning efforts, and merge all subsets to train tag models. By doing so, **the fine-tuned tag models can follow the system prompts strictly and generate different lengths of CoTs when given different system prompts in the inference stage.**
>
>
> **Q2:** Regarding the performance of QwQ-32B-Preview on the evaluation benchmarks when prompting the model with different reasoning efforts.
>
> **A2:** We also follow your suggestion to evaluate the performance of QwQ-32B-Preview on 3 evaluation benchmarks when prompting the model with different reasoning efforts. The evaluation parameters are the same as those used in our main experiments. **The following results are consistent with the results on the tag models and re-validate our main claim: longer CoTs may not necessarily lead to better performance.**
>
>
> Table 1: The performance of QwQ-32B-Preview on 3 benchmarks when prompting the model with different reasoning efforts.
>
> |Reasoning Effort|GSM8K||MATH500||AIME2024||
> |-|-|-|-|-|-|-|
> | |Acc.|#Tokens|Acc.|#Tokens|Acc.|#Tokens|
> |Low|**93.95**|418.26|**90.24**|1592.29|42.00|5152.97|
> |Medium|93.56|844.32| 89.16|2356.29|**44.00**|6413.97|
> |High|92.78| 1112.97|88.36|2684.87|41.33|6678.50|
>
> **Q3:** Regarding the comment "the ratios of erroneous reasoning rounds make more sense than the absolute number of erroneous reasoning rounds."
>
> **A3:** We follow your suggestion to display the ratios of erroneous reasoning rounds in responses under different reasoning efforts. As the following table shows, **the ratio of erroneous reasoning rounds also consistently increases as the reasoning effort increases**.
>
> Table 2: The statistics of responses under different reasoning efforts for training the tag models.
>
> |Reasoning Effort|Avg. Rounds|Avg. Erroneous Rounds|Ratio of Erroneous Rounds|
> |-|-|-|-|
> Low|1.64|0.11|0.067|
> Medium|2.28|0.20|0.088|
> High|2.81|0.32|0.114|
>
> **Q4:** Regarding additional experiments on supporting the claim that including more erroneous rounds in the training stage can lead to greater adverse effects.
>
> **A4:**  We follow your suggestion to filter out those solutions that contain erroneous steps (identified by GPT-4.1) under the high reasoning effort, and finetune Qwen2.5-32B-Instruct on the filtered data, which yields Qwen2.5-32B-Tag-High-Filtered.
>
> Table 3: The performance of the model trained on data with high reasoning effort after filtering out all solutions that are identified to contain erroneous steps.
>
> |Model|GSM8K||MATH500||AIME2024||
> |-|-|-|-|-|-|-|
> | |Acc. | #Tokens|Acc. | #Tokens|Acc. | #Tokens|
> |Qwen2.5-32B-Instruct (Temp=0.0)|95.91 |295.01 |84.20 |576.89 |16.67| 1407.43|
> |Qwen2.5-32B-Instruct (Temp=1.0)|95.30| 296.98| 82.84| 555.65|14.67|855.62|
> |Qwen2.5-32B-Tag-High|93.31|1820.17| 90.56| 3185.75|41.33|8753.87|
> |Qwen2.5-32B-Tag-High-Filtered|94.87 (+1.56)| 1478.10 (-342.07)|90.00 (-0.56)|2783.98 (-401.77)|36.33 (-5.00)|8049.29 (-704.58)|
>
>
> We have some interesting findings: (1) After removing samples containing erroneous rounds, the trained model produces much shorter CoTs. The reason is that the solutions removed are usually very lengthy as they contain more self-reflections and self-corrections, thus the average length of the dataset after removing these samples is much shorter. (2) Removing samples containing erroneous rounds brings significant performance improvement on GSM8K, while causing certain performance degradation on harder benchmarks MATH500 and AIME2024. The reason is that after removing those samples, the model cannot learn to effectively perform deeper thinking such as correcting the errors it could make in previous rounds. GSM8K is relatively simple, so the model does not need to engage in excessive reflection and correction. Therefore, removing these behaviors actually improves model performance. On more challenging datasets, the advantages of longer reasoning chains that include reflections and corrections become apparent. These findings are consistent with the main claim of our paper.
>
> Moreover, by comparing with the results in Figure 5 in the main text, we find that instead of directly removing correct solutions that contain erroneous steps, a more effective approach is to apply loss masking specifically to the wrong steps. The latter strategy allows the model to retain the ability to learn how to correct previous mistakes, without explicitly learning from the erroneous steps themselves.
>
>
> **Q5:** Regarding the question that "Why not just using the same data presented in Fig. 4 and masking out the loss with erroneous steps to demonstrate the point?"
>
> **A5:** The reason we do not use the responses for training tag models in the loss masking experiment is that **it is very difficult to accurately locate which specific steps are erroneous in those responses, due to the informal nature of o1-like outputs**. We have tried prompting LLMs to locate the exact erroneous steps, but found the results to be highly unreliable. In contrast, **when we construct our own long CoT data, we can precisely segment the reasoning traces step by step during the generation process, and then apply loss masking to the specific erroneous steps as needed**. We will include this clarification into the revision.
>
> **Q6:** Regarding future work on explorations in the RL setting.
>
> **A6:** **The primary scope and contribution of our work is to uncover the potential negative effects of scaling with excessively long chains of thought (CoTs).** Through comprehensive empirical analysis, we propose an efficient and effective self-improvement strategy, and demonstrate its effectiveness primarily in the SFT/DPO setting. We agree that **investigating the impact of CoT length in the RL setting is a valuable direction for future research, but it is out of the scope of this work**. As discussed in the Limitations section, our findings are also applicable to RL-based scaling. In particular, since RL typically assigns positive rewards (e.g., 1.0) to all solutions with correct final answers, it is preferable to favor shorter correct solutions with fewer erroneous reasoning steps over longer correct ones that contain more errors. Over-rewarding the latter may inadvertently encourage the model to generate incorrect intermediate steps and rely on its imperfect self-correction ability, rather than producing the correct answer directly.
>
> We hope the above clarifications address your concerns. We are glad to have further discussions if you have any further questions.

---

> > ### Comment · Reviewer_AhwA · 2025-08-02
> >
> > I thank the authors for addressing my concerns and questions.
> >
> > It is interesting to observe that the QwQ model performs worse when prompted to produce higher reasoning efforts. I suggest another potential explanation: this behavior might stem from poor alignment or inadequate instruction-following abilities. Alternatively, it could reflect a model-specific bias. It would be valuable to examine whether similar trends occur in other models.
> >
> > The fine-tuning results on filtered data are very interesting. Recent research [1] suggests that the correctness of reasoning traces might not significantly impact performance, as incorrect reasoning traces in SFT can yield similar results to correct ones. This appears to contradicts with the findings presented here. Could the authors provide commentary on this? Additionally, what if one uses a stronger model to correct the erroneous steps in the reasoning traces of high-reasoning effort and fine-tune on this data? Would this help? This experiment can further strengthen and support that authors claim that the erroneous steps are the main reason for performance degradation.
> >
> > Overall, I find the paper to be of high quality, and I intend to raise my score to 5.
> >
> > [1] Du, Wei, et al. "The Challenge of Teaching Reasoning to LLMs Without RL or Distillation." 2nd AI for Math Workshop @ ICML 2025. 2025.

---

> ### Author Response · Authors · 2025-08-03
> **Thank you**
>
> We sincerely thank you for your positive feedback and strong support for our work!
>
> To further investigate whether prompting models to generate longer responses also leads to decreased performance in other models, we conducted additional experiments on QwQ-32B. Since QwQ-32B includes both reasoning and summary components in its responses and typically generates much longer CoTs, we extended its max_seq_len to 32K to allow for sufficient reasoning and summarization. The results (averaged on 5 samplings) are presented in the following table. As shown, **same trends are observed in this more advanced model**: higher reasoning effort does not always yield better performance. However, we agree with your point that one possible reason is that current reasoning models may have inadequate instruction-following abilities and tend to overfit to the training prompt template, as you mentioned. Our reasoning effort-based fine-tuning experiments can help mitigate this issue by directly aligning the model’s response length distribution with specific reasoning efforts.
>
> Table 1: The performance of QwQ-32B when directly prompting the model with different reasoning efforts.
>
> |Reasoning Effort|GSM8K||MATH500||AIME2024||
> |-|-|-|-|-|-|-|
> | |Acc.|#Tokens|Acc.|#Tokens|Acc.|#Tokens|
> |Low|**96.56**|568.49|95.76|2532.82|74.00|11390.70|
> |Medium|96.50|1152.86| **96.08**| 3708.79|78.00|12969.53|
> |High|96.36|1752.29|95.80 | 4198.67 |**80.67** |13194.29|
>
>
> Regarding the results in [1], which show that using incorrect responses in SFT can yield similar outcomes to using correct ones, we have two main explanations: (1) The SFT dataset in [1] is very small (only 50 samples), which allows the model to quickly overfit to the o1-like long reasoning trace format and generalize this reasoning style to other problems. In this scenario, the correctness of the reasoning chain has a relatively smaller impact on generalization compared to the format itself. (2) Another possible reason is data contamination in the Qwen2.5 model. Recent work [2] finds that even using incorrect rewards for RL on Qwen2.5 can also effectively improve model performance, and [3] suggests that this may be due to data contamination in the Qwen2.5 training data. We believe this could also partially explain the experimental findings in [1].
>
>
> Regarding your final question about using a stronger model to correct the erroneous steps in the reasoning traces of high reasoning effort, we would like to clarify that: If the corrected steps are simply appended after the wrong steps as error correction steps, but loss masking is not applied to the erroneous steps during SFT, then both the experimental setup and results are essentially the same as those shown in our Figure 5 in the main paper. On the other hand, if the wrong steps are discarded and replaced directly with the corrected steps, this is equivalent to the experimental setup and results on our filtered high reasoning effort-based set (see Table 3 in our rebuttal).
>
> We sincerely thank you for your insightful questions again, your support means a lot to us! We are glad to have further discussions with you if you have any follow-up questions.
>
> [1] Du, Wei, et al. "The Challenge of Teaching Reasoning to LLMs Without RL or Distillation." 2nd AI for Math Workshop @ ICML 2025. 2025.
>
> [2] Shao, Rulin, et al. "Spurious rewards: Rethinking training signals in rlvr." arxiv 2025
>
> [3] Wu, Mingqi, et al. "Reasoning or Memorization? Unreliable Results of Reinforcement Learning Due to Data Contamination." arxiv 2025

---

> > ### Comment · Reviewer_AhwA · 2025-08-04
> >
> > I thank the authors for providing additional results. I have raised my score to 5 in reflection of the improvement the authors have made.

---

> > > ### Author Response · Authors · 2025-08-04
> > > **Thank you for your support**
> > >
> > > We sincerely thank you again for your great support! We will incorporate your feedback into the revision.

---

### Official Review · Reviewer_EPhU · 2025-07-05

**Clarity:** 3
**Significance:** 4
**Originality:** 3
**Rating:** 4
**Confidence:** 3

**Summary:**

This works makes a very important and timely observation that if an LLM can already answer a question correctly under the given reasoning effort, increasing the response length with additional tokens may have adverse effects as longer responses are more likely to include erroneous steps. However, encouraging LLMs to think more is beneficial to tackle more challenging problems. This work proposes a method that use a small set of o1-like responses under different reasoning efforts (i.e., of varying lengths) to train a “tag” model, which
is used to generate responses for a large set of math problems under different reasoning efforts. Then, it selects the shortest correct response generated across all reasoning efforts to create a thinking-optimal dataset, which is used for the self-improvement of the base
model.

**Questions:**

1. Only three math data is provided for reasoning task. Will the observation hold for other reasoning tasks like commonsense reasoning, coding etc ?
2. Will the method work for these new reasoning task ?

**Ethical Concerns:**

["NO or VERY MINOR ethics concerns only"]

**Final Justification:**

I think the paper is timely and present interesting insights for the community. The rebuttal also addressed all my issues. However after looking at other rebuttal, I decided to keep my score.

**Quality:**

4

**Strengths And Weaknesses:**

Strengths:

1. The work is very insightful and relevant to the community.
2. The work is very well motivated with analysis on state of the art LLM models.
3. The insights provided on "the impact of scaling efforts on the effectiveness of test-time scaling" is very useful, timely and a big novelty of the this.

Weakness:

1. The improvement is not a lot. Hence it's important to see statistical significance results. Both for table 6 and table 2.
2. 3 dataset are not enough to evaluate.
3. Only math reasoning task is studied.
4. The method is not backed by theory or a justification/intuition on how the method works.

---

> ### Author Rebuttal · Authors · 2025-07-31
>
> We sincerely thank you for your positive review and your constructive comments. We are glad that you think our paper is well-motivated and very insightful. We are encouraged that you find our insights very useful. We make the following response to address your remaining concerns.
>
> **Q1:** Regarding the statistical significance results.
>
> **A1:** We have displayed the statistical significance results of Table 2 (self-improvement results on Qwen2.5-32B-Instruct) in Table 7 in Appendix I. Here, we show the standard deviation results of Table 6 (self-improvement results on LLaMA3.1-8B-Instuct) in the following table for your reference. We will add it into the revision.
>
> Table 1: Standard deviation results of self-improvement results on LLaMA3.1-8B-Instuct.
>
> |Model|GSM8K|MATH500|AIME2024|
> |-|-|-|-|
> |LLaMA3.1-8B-Instruct (Temp=0.0)|82.18|47.00|6.67|
> |LLaMA3.1-8B-Instruct (Temp=1.0)|76.21(±0.66)|39.60(±0.84)|4.67(±2.98)|
> |LLaMA3.1-8B-Random|87.94(±0.33)|60.52(±1.38)|4.67(±1.83)|
> |LLaMA3.1-8B-TOPS (ours)|88.54(±0.26)|61.28(±0.73)|8.00(±1.83)|
>
> **Q2:** Regarding the observations on more datasets and other reasoning tasks.
>
> **A2:** We follow your suggestion to perform the same empirical validation in the general reasoning domain as we did in Section 3.2. Specifically, we prompt QwQ-32B-Preview to generate responses under different reasoning effort-based prompts on a subset of the WebInstruct-verified dataset [1], and curate a seed general reasoning dataset that contains 3.6K correct responses with varying reasoning efforts. We then finetune Qwen2.5-7B-Instruct on the seed dataset to get Qwen2.5-7B-Tag-General, and evaluate its performance on MMLU-Pro and GPQA-Diamond using the same reasoning effort-conditioned prompting strategy as in Section 3.2 (only sample once for each prompt and set max_seq_length to 8K considering the huge evaluation cost). The results are displayed in the following table. **The findings are consistent with the observations in the math reasoning domain that excessive scaling with longer CoTs can bring negative effects to the model’s performance in general reasoning tasks.** Also, we can find that the model needs more reasoning effort to perform well on the more challenging task GPQA-Diamond.
>
> Table 2: The performance of Qwen2.5-7B-Tag-General on MMLU-Pro and GPQA-Diamond.
>
> |Model|MMLU-Pro||GPQA-Diamond||
> |-|-|-|-|-|
> | |Acc. | #Tokens|Acc. | #Tokens|
> |Qwen2.5-7B-Instruct (Temp=0.0)|52.46|401.38|34.85|592.73|
> |Qwen2.5-7B-Instruct (Temp=1.0)|51.49|379.84|33.84|537.41|
> |Qwen2.5-7B-Tag-General-Low|**56.00**|1674.60|31.82 | 2808.88|
> |Qwen2.5-7B-Tag-General-Medium|55.92|2341.27|**36.87**|3931.13|
> |Qwen2.5-7B-Tag-General-High|55.81|2632.05|32.83|4238.11|
>
>
>
>
> **Q3:** Regarding the performance of TOPS on other reasoning tasks.
>
> **A3:** Following the setup in A2, we then perform our thinking-optimal scaling strategy on a held-out set of the WebInstruct-verified dataset to get Qwen2.5-7B-TOPS-General and perform random scaling to get Qwen2.5-7B-Random-General. The evaluation results in the following table show that **our method can also work well for general reasoning tasks**.
>
> Table 3: The self-improvement results on MMLU-Pro and GPQA-Diamond.
>
> |Model|MMLU-Pro||GPQA-Diamond||
> |-|-|-|-|-|
> | |Acc. | #Tokens|Acc. | #Tokens|
> |Qwen2.5-7B-Instruct (Temp=0.0)|52.46|401.38|34.85|592.73|
> |Qwen2.5-7B-Instruct (Temp=1.0)|51.49|379.84|33.84|537.41|
> |Qwen2.5-7B-Random-General|55.86|1960.87|34.34|3385.70|
> |Qwen2.5-7B-TOPS-General (ours)|**56.50**|1788.74|**38.38**|3083.76|
>
>
> **Q4:** Regarding the intuition on how the method works.
>
> **A4:** **Our method is backed by the empirical analysis in Section 3.2 and Section 3.3.** As shown in Figure 2 and Figure 3, even though all the trainings are performed on solutions that are all ultimately correct, scaling with longer CoTs can adversely affect the model’s reasoning performance in certain domains, and there exists an optimal reasoning effort that varies across different tasks of varying difficulty levels. Figure 4 and Figure 5 help to explain such phenomenon: longer responses are more likely to contain more erroneous steps, and excessive training on such responses can degrade the model’s reasoning performance. Based on these intuitions, we propose to create a thinking-optimal dataset that contains the shortest correct response of the model on each prompt across all reasoning efforts, because this is the most efficient and effective trace that needs to be learned.
>
>
> We hope the above clarifications address your concerns. We are glad to have further discussions if you have any further questions.
>
> [1] Ma, Xueguang, et al. "General-reasoner: Advancing llm reasoning across all domains." arxiv 2025

---

> > ### Author Response · Authors · 2025-08-05
> > **Looking forward to your feedback**
> >
> > Dear Review EPhU,
> >
> > We sincerely thank you again for your positive reviews and constructive comments on our paper. We have followed your suggestions to provide the standard deviation results, the additional experiments on general reasoning tasks and the performance of our method on new tasks. We have also provided a discussion on the intuition on how our method works. As the author-reviewer discussion deadline is approaching, we would like to know if you have any further questions, and we are glad to continue the discussion if there are any.
> >
> > We are sincerely looking forward to your feedback, and your support means a lot to us! Thank you.
> >
> > Best regards,
> >
> > Authors

---

> > > ### Comment · Reviewer_EPhU · 2025-08-07
> > >
> > > Thanks for the nice rebuttal. I have decided to maintain my score.

---

> > > > ### Author Response · Authors · 2025-08-07
> > > > **Thank you for your support**
> > > >
> > > > We sincerely thank you for your positive feedback and your great support for our work! We are glad that our responses have addressed all your questions. We will incorporate your feedback into the revision.

---

### Official Review · Reviewer_CZmU · 2025-07-10

**Clarity:** 3
**Significance:** 2
**Originality:** 2
**Rating:** 3
**Confidence:** 4

**Summary:**

- This paper presents an analysis and a new algorithm to modulate the amount of reasoning effort (length of CoT) for language models.
 - First, the paper studies the observation that training on longer reasoning paths hurts accuracy on easier problems because empirically (as judged by GPT-4o), additional reasoning processes increaess the probability of error.
 - Then the authors propose a strategy, TOPS, that varies the amount of reasoning effort based on the problem. In particular, they train a model to condition on the amount of reasoning effort, and then learna model to allocate fewer tokens to easier problems and more tokens to harder problems (for each problem, they determine the shortest reasoning trace that yields the correct answer).
 - In the end, they produce a model that performs comparably to QwQ-32B-Preview on GSM8K, MATH500, AIME2024.

**Questions:**

- Figure 4 shows that the number of erroneous reasoning steps increases with length. What's the mapping from this to whether the final answer is correct or not? Is it that one erroneous step is enough to kill the solution or is there some robustness (e.g., at least two have to be wrong)? Fundamentally though, it seems like the metric at the end is more of an "and" rather than a "sum" (almost everything has to be right).
 - How are the length distributions of the reasoning effort-conditioned model? Does the model actually generate reasoning chains of the appropriate length?
 - I would have liked to see a bit more qualitative discussion of what modulating the reasoning effort does to the reasoning chain? Does the model just skip steps if asked to make it shorter or are there fundamentally different strategies?
 - As one changes the amount of reasoning effort, what happens to the distribution of answers (I think it's always helpful to think about the whole distribution, acknowledging that there is an increased computation cost)?

**Ethical Concerns:**

["NO or VERY MINOR ethics concerns only"]

**Final Justification:**

I increased my score by 1 based on the rebuttal.

**Limitations:**

yes

**Quality:**

3

**Strengths And Weaknesses:**

- Trying to get models that vary the amount of reasoning effort based on the problem is a very sensible goal.
 - Training a model to condition on the amount of reasoning effort is good too (this reduces the number of potential confounders).
 - Using the shortest one reasoning trace seems problematic conceptually. For each amount of reasoning effort, one really has a probability of correctness. So then what does the "shortest reasoning trace" mean? Practically speaking, the dataset that's collected by (2) might be quite noisy.
 - The actual accuracies are not particularly noteworthy (AIME2024 is in the 40s, but there are 32B models now that are in the 70s - e.g., QwQ-32B).

---

> ### Author Rebuttal · Authors · 2025-07-31
>
> We sincerely thank you for your great efforts on reviewing our paper and your helpful questions. We are glad that you think our target is a sensible goal and our method is good. We make the following response to address your questions.
>
> **Q1:** Regarding the question "what does the "shortest reasoning trace" mean" and the comment "the dataset that's collected by (2) might be quite noisy".
>
> **A1:** The "shortest reasoning trace" refers to the **correct solution with the minimal response length that can be generated by the target model across all levels of reasoning effort**. This trace represents the most efficient path the model can take to reach the correct final answer. As analyzed in Section 3.3, other longer responses are more likely to contain additional erroneous steps, even if their final answers are correct, and training on such responses can degrade model performance.
>
> Based on these findings, we propose constructing thinking-optimal datasets using Eq. (2) to enable more effective test-time scaling. We agree with you that datasets collected via Eq. (2) may be noisy, because it is nontrivial to precisely define and find the truly shortest correct response. However, we believe that increasing the number of samples for each prompt under each reasoning effort can help mitigate this noise and improve the reliability of the thinking-optimal dataset.
>
>
> **Q2:** Regarding the comment "the actual accuracies are not particularly noteworthy".
>
> **A2:** **The main scope of our work is to reveal the potential side effects of scaling with excessively long CoTs and to propose a more efficient and effective self-improvement strategy.** Our self-improved model achieves comparable performance to QwQ-32B-Preview with more efficient and adaptive thinking modes using only 1.3K seed data, and consistently outperforms other direct distillation-based models which rely on more high-quality distillation data in both effectiveness and efficiency. Also, our thinking-optimal model significantly outperforms random scaling (Qwen2.5-32B-Random) under the same constraints, validating our initial motivation. Finally, we believe **our method is also applicable when using other models such as QwQ-32B to curate seed data for self-improvement**.
>
> **Q3:** Regarding the question "What's the mapping from the number of erroneous reasoning steps to whether the final answer is correct or not?"
>
> **A3:** **For Figure 4, all candidate responses have correct final answers.** The purpose of Figure 4 and Figure 5 is to elucidate the empirical findings presented in Figure 2 and Figure 3: although all training solutions ultimately yield correct answers, scaling with longer CoTs can adversely affect the model’s reasoning performance in certain domains. This is because longer CoTs are more likely to include additional erroneous reasoning steps, and excessive training on such responses may degrade the model’s overall reasoning ability.
>
> **Q4:** Regarding the length distributions of the reasoning effort-conditioned model.
>
> **A4:** **Figure 3 displays the length distributions of the reasoning effort-conditioned models (i.e., tag models).** As we can see, the reasoning effort-conditioned models achieve the best reasoning performance on GSM8K under the low reasoning effort with fewer than 500 average tokens, and achieve promising performance on AIME2024 under the high reasoning effort with more than 8000 average tokens. **This helps to validate that the reasoning effort-conditioned models can actually generate reasoning chains of the appropriate lengths.**
>
> **Q5:** Regarding the qualitative discussion of what modulating the reasoning effort does to the reasoning chain.
>
> **A5:** Thank you for your constructive question. We follow your suggestion to manually analyze the reasoning chains under different reasoning efforts. We find that in the vast majority of cases, responses under low reasoning efforts use fewer complete reasoning rounds compared to responses under higher reasoning efforts (consistent with Figure 4). This property manifests differently in simple and difficult problems:
>
> (1) For simple problems (about 40% samples we checked), higher reasoning effort leads to **overthinking**, where the model continuously adopts alternative strategies to re-solve the problem. This causes the model to initially answer correctly but subsequently obtain incorrect answers through re-solving, and then become misled by these wrong answers, ultimately leading to incorrect responses.
>
> (2) For some difficult problems, the property of engaging in continuous reflection, correction, and exploration under higher reasoning efforts enables the model to solve problems that cannot be addressed by lower reasoning efforts due to **underthinking**.
>
> **Q6:** Regarding the question "As one changes the amount of reasoning effort, what happens to the distribution of answers?"
>
> **A6:** We follow your suggestion to calculate the distributions of answers of reasoning effort-conditioned models under different reasoning efforts. Specifically, we calculate the average number of distinct answers in 5 samples per prompt under each reasoning effort. We also display the accuracy on each benchmark in the following table for reference. We find some very interesting phenomena: **Generally, the reasoning effort that achieves the best performance on a certain benchmark also leads to the lowest average number of distinct answers per prompt.** This indicates that, **under the optimal thinking effort, the model can generate the most consistent answers across multiple samplings without either underthinking or overthinking.** Your question is helpful for better understanding of empirical findings in Section 3.2 and we will add it into the revision.
>
> Table 1: The distribution of answers under different reasoning efforts along with the average accuracy. We highlight the best accuracy and the lowest average number of distinct answers per prompt.
>
> |Reasoning Effort|GSM8K||MATH500||AIME2024||
> |-|-|-|-|-|-|-|
> | | Avg. # Distinct Answers | Accuracy |Avg. # Distinct Answers | Accuracy |Avg. # Distinct Answers | Accuracy |
> |***Qwen Models***|
> |Qwen2.5-32B-Tag-Low|**1.13**| **95.53** | 1.39| 90.60 | 3.20|34.67 |
> |Qwen2.5-32B-Tag-Medium|1.14|94.33 |**1.36** | **91.48** | **3.13** | **42.00** |
> |Qwen2.5-32B-Tag-High|1.15| 93.31 | 1.40| 90.56 | 3.30| 41.33|
> |***LLaMA Models***|
> |LLaMA3.1-8B-Tag-Low|**1.37**| **87.26** | **2.34** | 59.00 |4.27|7.33|
> |LLaMA3.1-8B-Tag-Medium|1.42|87.06 | 2.54| **61.12** |4.27|7.33 |
> |LLaMA3.1-8B-Tag-High|1.47| 86.89| 2.59 |  59.36 |**4.00**| **10.00**|
>
> We hope the above clarifications address your concerns. We are glad to have further discussions if you have any further questions.

---

> ### Author Response · Authors · 2025-08-05
> **Looking forward to your feedback**
>
> Dear Review CZmU,
>
> We sincerely thank you again for reviewing our paper and providing helpful comments. We have addressed your questions about the shortest correct response, the self-improvement performance and the correctness of final answers in reasoning effort-conditioned training data in the rebuttal. Following your suggestion, we have included the discussion on the effects of reasoning efforts on reasoning traces, and provided an interesting empirical analysis on the answer distributions under different reasoning efforts. As the author-reviewer discussion deadline is approaching, we would like to know if you have any further questions, and we are glad to continue the discussion if there are any.
>
> We are sincerely looking forward to your feedback, and your support means a lot to us! Thank you.
>
> Best regards,
>
> Authors

---

> > ### Author Response · Authors · 2025-08-06
> > **Sincerely looking forward to your feedback**
> >
> > Dear Review CZmU,
> >
> > We hope this message finds you well. We have addressed all your concerns in our rebuttal. It has been six days since our response, and we sincerely hope to hear your feedback. We hope you are satisfied with our rebuttal, your support means a lot to us! If you have any further questions, we would be happy to continue the discussion. Thank you.
> >
> > Best regards,
> >
> > Authors

---

> > > ### Author Response · Authors · 2025-08-07
> > > **Sincerely looking forward to your feedback**
> > >
> > > Dear Review CZmU,
> > >
> > > We hope this message finds you well. Since the author-reviewer discussion will end within two days, we sincerely hope to hear your feedback. If you have any further questions, we are happy to continue the discussion. Thank you again.
> > >
> > > Best regards,
> > >
> > > Authors

---

> > > > ### Comment · Reviewer_CZmU · 2025-08-08
> > > >
> > > > Thank you for the detailed rebuttal and also running the new experiment.  I still think that the relationship between reasoning length and accuracy is a delicate one that one should rigorously explore more.  How generalizable these results are to other datasets and models is an open question, since I'm generally a bit worried that a lot of the reasoning research is overfit to math and Qwen models.  But in the end, I think there are some interesting ideas here, and I have increased my score by 1.  I hope the authors will continue to dig deeper and improve the paper.

---

> ### Author Response · Authors · 2025-08-08
> **Additional results on other tasks and models**
>
> Dear Reviewer CZmU,
>
> We sincerely thank you for your positive feedback. **We have conducted additional experiments on other datasets (general reasoning tasks) and models (LLaMA).** We put the new results in the following for your reference.
>
> First, we put the self-improvement results of our method on LLaMA models in Table 1. **As we can see, our method can also work well on other models.**
>
> Table 1: Self-improvement results on LLaMA3.1-8B-Instuct.
>
> |Model|GSM8K|MATH500|AIME2024|
> |-|-|-|-|
> |LLaMA3.1-8B-Instruct (Temp=0.0)|82.18|47.00|6.67|
> |LLaMA3.1-8B-Instruct (Temp=1.0)|76.21|39.60|4.67|
> |LLaMA3.1-8B-Random|87.94|60.52|4.67|
> |LLaMA3.1-8B-TOPS (ours)|**88.54**|**61.28**|**8.00**|
>
> Additionally, we prompt QwQ-32B-Preview to generate responses under different reasoning effort-based prompts on a subset of the WebInstruct-verified dataset [1], and curate a seed general reasoning dataset that contains correct responses with varying reasoning efforts. We then finetune Qwen2.5-7B-Instruct on the seed dataset to get Qwen2.5-7B-Tag-General, and evaluate its performance on MMLU-Pro and GPQA-Diamond using the same reasoning effort-conditioned prompting strategy as in Section 3.2 (only sampling once for each prompt considering the evaluation cost). The results are displayed in the following Table 2. **The findings are consistent with the results in the math reasoning domain that excessive scaling with longer CoTs can bring negative effects to the model’s performance in general reasoning tasks.** Also, the model needs more reasoning effort to perform well on the more challenging task GPQA-Diamond.
>
> Table 2: The performance of Qwen2.5-7B-Tag-General on MMLU-Pro and GPQA-Diamond.
>
> |Model|MMLU-Pro||GPQA-Diamond||
> |-|-|-|-|-|
> | |Acc. | #Tokens|Acc. | #Tokens|
> |Qwen2.5-7B-Instruct (Temp=0.0)|52.46|401.38|34.85|592.73|
> |Qwen2.5-7B-Instruct (Temp=1.0)|51.49|379.84|33.84|537.41|
> |Qwen2.5-7B-Tag-General-Low|**56.00**|1674.60|31.82 | 2808.88|
> |Qwen2.5-7B-Tag-General-Medium|55.92|2341.27|**36.87**|3931.13|
> |Qwen2.5-7B-Tag-General-High|55.81|2632.05|32.83|4238.11|
>
>
>
> Following the above setup, we then perform our thinking-optimal scaling on a held-out set of the WebInstruct-verified dataset to get Qwen2.5-7B-TOPS-General and perform random scaling to get Qwen2.5-7B-Random-General. The evaluation results in the following table show that **our method can also work well for general reasoning tasks**.
>
> Table 3: The self-improvement results on MMLU-Pro and GPQA-Diamond.
>
> |Model|MMLU-Pro||GPQA-Diamond||
> |-|-|-|-|-|
> | |Acc. | #Tokens|Acc. | #Tokens|
> |Qwen2.5-7B-Instruct (Temp=0.0)|52.46|401.38|34.85|592.73|
> |Qwen2.5-7B-Instruct (Temp=1.0)|51.49|379.84|33.84|537.41|
> |Qwen2.5-7B-Random-General|55.86|1960.87|34.34|3385.70|
> |Qwen2.5-7B-TOPS-General (ours)|**56.50**|1788.74|**38.38**|3083.76|
>
> **We sincerely hope the above results address your last question well, and you can provide more support if you find these new results helpful!** Your support means a lot to us! Thank you again.
>
> Best regards,
>
> Authors
>
> [1] Ma, Xueguang, et al. "General-reasoner: Advancing llm reasoning across all domains." arxiv 2025

---

> > ### Author Response · Authors · 2025-08-08
> > **Regarding your last comment**
> >
> > Dear Reviewer  CZmU,
> >
> > We thank you very much for your latest feedback! **We have attached the results on additional tasks and models to address you last question about the generalizability of our observations and method, which are also acknowledged by Reviewer EPhU and Reviewer B8YC.** We hope you can check our latest response, and we would be very glad to receive your greater support if you are satisfied with the new results. Thank you again.
> >
> > Best regards,
> >
> > Authors

---

### Author Response · Authors · 2025-08-09
**Thank you for your efforts!**

Dear all Reviewers, ACs, and SACs,

We sincerely thank you for your precious time and your constructive reviews on improving our manuscript! **Our paper presents the novel and unique contribution to uncover the potential negative effects of scaling with excessively long chains of thought (CoTs).** We conduct extensive experiments and analysis to support this finding, and based on this, we propose a Thinking-Optimal Scaling method to help LLMs achieve more effective and efficient self-improvement on deep reasoning.

We are encouraged by the great recognition from the reviewers, which includes

- Well-motivated and sensible research goal and good method (Reviewer ```CZmU```, Reviewer ```EPhU```)
- Novel insights and interesting findings that contribute to the understanding of test-time scaling (Reviewer ```CZmU```, Reviewer ```EPhU```,  Reviewer ```AhwA```)
- Thorough empirical evaluations showcasing the method's effectiveness (Reviewer ```B8YC```)
- Well-written paper with an easy-to-understand structure (Reviewer ```AhwA``` and Reviewer ```B8YC```)

During author-reviewer discussion, we have conducted following additional experiments to address all the reviewers' concerns:

- We **conduct additional experiments in general reasoning domain** to demonstrate generalizability of our observations and the effectiveness of method in other tasks. (Reviewer ```CZmU```, Reviewer ```EPhU```, Reviewer ```B8YC```)

- We **display the self-improvement results on LLaMA3.1-8B-Instruct** to demonstrate the generalizability of our method on non-Qwen models. (Reviewer ```CZmU```, Reviewer ```EPhU```, Reviewer ```B8YC```)

- We **calculate the distributions of answers of reasoning effort-conditioned models under different reasoning efforts** and find that under the optimal thinking effort, the model can generate the most consistent answers across multiple samplings without either underthinking or overthinking, which provides deeper understanding of our main findings (Reviewer ```CZmU```)

- We **show the performance of QwQ-32B-Preview  and QwQ-32B on several benchmarks when prompting the model with different reasoning efforts** to re-validate our main claim that longer CoTs may not necessarily lead to better performance (Reviewer ```AhwA```)

- We **conduct experiments to filter out seed solutions that contain erroneous steps under the high reasoning effort, and finetune Qwen2.5-32B-Instruct on the filtered data** to further support our claim that including more erroneous rounds in the training stage can lead to greater adverse effects (Reviewer ```AhwA```)

We are glad that our responses and new experiments have addressed all reviewer's questions (the additional experiments on general reasoning tasks and LLaMA models address the last comment of Reviewer ```CZmU```) and we are encouraged to receive positive feedback from all reviewers. We will incoporate these helpful feedback and discussions into the revision.

Thank you very much agian!

Best regards,

Authors of Submission 8726

---

### Decision · Program_Chairs · 2025-09-17

**Decision:**

Accept (poster)

**Comment:**

This paper focus on inference time compute optimization. It shows that overly long reasoning paths on easy problems can actually hurt the model performance. Motivated by this, it proposed a strategy of adaptively selecting optimal thining budget to address this issue. The proposed method is shown to be effective at reducing overthinking on easy problems while maintaining the effectiveness on difficult problems.

The research topic is timely given the current trends of scaling test time compute in LLMs. However, the paper received borderline ratings due to some limitations. For examples, there are some arbitrary design choices such as using the "shortest reasoning traces" that could benefits from more analysis; and it would be useful to try to experiment on baseline models with close-to-SoTA performance on some dataset such as AIME24. Nethertheless, this paper has improved noticeably during the rebuttal with additional results and clarifications that could be improved. We generally agreed that the paper is a valuable contribution to the field despite some of the limitations and recommend acceptance. We recommend the authors to still try to address some of the questions and suggestions raised by the reviewers in the final revision to produce a solid publication.